# Yap-lin28a axis targets let7-Wnt pathway to restore progenitors for initiating regeneration

**Zhian Ye[1†], Zhongwu Su[2†], Siyu Xie[1,3], Yuye Liu[1], Yongqiang Wang[1], Xi Xu[1,3], Yiqing Zheng[2*], Meng Zhao[3*], Linjia Jiang[1*]**

[1]Guangdong Provincial Key Laboratory of Malignant Tumor Epigenetics and Gene Regulation, Sun Yat-Sen Memorial Hospital, Sun Yat-Sen University, Guangzhou, China; [2]Department of Otolaryngology, Sun Yat-Sen Memorial Hospital, Sun Yat-Sen University, Guangzhou, China; [3]Key Laboratory of Stem Cells and Tissue Engineering, Zhongshan School of Medicine, Sun Yat-Sen University, Ministry of Education, Guangzhou, China

**Abstract** The sox2 expressing (sox2[+]) progenitors in adult mammalian inner ear lose the capacity to regenerate while progenitors in the zebrafish lateral line are able to proliferate and regenerate damaged HCs throughout lifetime. To mimic the HC damage in mammals, we have established a zebrafish severe injury model to eliminate both progenitors and HCs. The *atoh1a* expressing (*atoh1a[+]*) HC precursors were the main population that survived post severe injury, and gained sox2 expression to initiate progenitor regeneration. In response to severe injury, *yap* was activated to upregulate *lin28a* transcription. Severe-injury-induced progenitor regeneration was disabled in *lin28a* or *yap* mutants. In contrary, overexpression of *lin28a* initiated the recovery of sox2[+] progenitors. Mechanistically, microRNA *let7* acted downstream of *lin28a* to activate Wnt pathway for promoting regeneration. Our findings that lin28a is necessary and sufficient to regenerate the exhausted sox2[+] progenitors shed light on restoration of progenitors to initiate HC regeneration in mammals.

**\*For correspondence:**
zhengyiq@mail.sysu.edu.cn (YZ);
zhaom38@mail.sysu.edu.cn (MZ);
jianglj7@mail.sysu.edu.cn (LJ)

[†]These authors contributed equally to this work

## Introduction

The auditory epithelium is a delicate structure located in the cochlea that is composed of sensory hair cells (HCs) and nonsensory support cells (SCs). During early development of mouse cochlea, the transcription factor sox2 is required to determine the prosensory region that mainly contains progenitors (*Julian et al., 2013*; *Kiernan et al., 2005*; *Dabdoub et al., 2008*). From E12.5 to E14.5 sox2[+] progenitors exit cell cycle and differentiate into auditory HCs and SCs. Basic helix-loop-helix (bHLH) transcription factor atoh1 acts as the cardinal gene initiating auditory HC differentiation since atoh1 deficiency causes complete loss of cochlear HCs (*Bermingham et al., 1999*). A subset of post-mitotic progenitors start to express high levels of atoh1 and downregulate sox2 expression to differentiate into the HC precursors (*Dabdoub et al., 2008*; *Zhong et al., 2019*; *Cai and Groves, 2015*; *Zhang et al., 2017*). Afterwards, their terminal differentiation toward mature HCs is promoted by the upregulation of atoh1 target genes, such as pou4f3 (*Cai et al., 2015*). In the meanwhile, differentiating HCs secret Notch ligands to activate Notch pathway, which inhibits atoh1 expression in neighboring cells and forces them to adopt the SC fate (*Abdolazimi et al., 2016*; *Costa et al., 2017*; *Lanford et al., 1999*).

The zebrafish lateral line is a mechanosensory organ composed of a series of neuromasts distributed on the body surface for detecting water flow. The lateral line HCs share similarities with their counterparts in mammalian inner ear in morphology, function and developmental pathways

(*Whitfield, 2002*; *Nicolson, 2005*). Each neuromast contains sensory HCs in the center surrounded by SCs and mantle cells (MCs). Aminoglycoside antibiotics, such as neomycin, ablates mature HCs and initiates robust mitotic regeneration that is characterized with SC division and differentiation (*Harris et al., 2003*; *Jiang et al., 2014*; *Williams and Holder, 2000*; *Ma et al., 2008*; *Romero-Carvajal et al., 2015*). The powerful capacity to regenerate HCs sustains after multiple rounds of damage, and retains throughout lifetime (*Cruz et al., 2015*; *Pinto-Teixeira et al., 2015*). Even after severe loss of the tissue integrity, the residual SCs have high potential to recover the neuromast by acting plastic to generate all three cell types (*Viader-Llargués et al., 2018*). Like in inner ear, *atoh1a* is expressed in HC precursors but not mature HCs in neuromast while *sox2* is expressed in a part of SCs and MCs (*Ma et al., 2008*; *Lush et al., 2019*). Sox2[+] SCs behave as progenitors to proliferate and differentiate through activation of canonical Wnt pathway during regeneration (*Hernández et al., 2007*; *Jacques et al., 2014*). However, it is unknown how regeneration is initiated when sox2[+] progenitors are absent.

Mammalian sensory HCs are vulnerable to damages caused by antibiotics, chemotherapeutical drugs and noise, which results in various hearing and balance diseases (*Cox et al., 2014*). Until now, the principal method used to initiate auditory HC regeneration in mammalian inner ear is to induce the transdifferentiation of SCs into HCs by upregulating atoh1 expression. For example, many studies tried to overexpress atoh1 in SCs with adenovirus, or used Notch inhibitor to increase atoh1 expression (*Atkinson et al., 2018*; *Mizutari et al., 2013*; *Izumikawa et al., 2005*; *Yang et al., 2012*). However, because the efficiency of HC induction is very low and SCs are lost due to transdifferentiation, very limited progress toward hearing recovery has been achieved (*Cox et al., 2014*; *Zheng and Zuo, 2017*; *Chen et al., 2019*). New strategies of restoring sox2[+] progenitors to initiate mitotic regeneration would be more promising to realize functional regeneration in mammalian adult inner ear. Unfortunately, very little is known whether and how sox2[+] progenitors can be restored in sensory epithelium.

Here in the zebrafish lateral line, we found that exhausted sox2[+] progenitors were able to restore quickly for initiating HC regeneration in severe injury. Atoh1a[+] HC precursors were the main population that survived post severe injury and dedifferentiated into *sox2*[+] progenitors through yap-lin28a pathway.

## Results

### Exhausted sox2[+] progenitors were quickly recovered by intensive proliferation post severe injury

It is well documented that Notch signaling pathway negatively regulates the differentiation of SCs into HCs (*Ma et al., 2008*; *Haddon et al., 1998*). Using the γ-secretase inhibitor LY411575 to inhibit Notch pathway from 3-day-post-fertilization (dpf) to 5dpf, we found that SCs were exhausted by persistent differentiation into HCs (*Figure 1—figure supplement 1*). Neomycin was used to ablate mature HCs following LY treatment (LY+neo), which leads to the severe injury with both HCs and SCs being ablated. Sox2 marks the proliferative progenitors that produce both SCs and HCs in homeostatic and regenerative neuromasts (*Hernández et al., 2007*). We analyzed sox2[+] cells in neo- and LY+neo-treated neuromast by immunostaining with anti-sox2 antibody. While the distribution of sox2[+] cells moved to the center post neo compared with normal larvae, the number was not changed. But sox2[+] cell number was dramatically decreased after LY or LY+neo (*Figure 1A–B*). Surprisingly, sox2[+] progenitors were able to recover quickly post LY+neo. A few sox2[+] progenitors appeared early at 6 hr, with increased sox2[+] progenitors being regenerated afterwards and recovered to normal level by 48 hr. In *Figure 1A*, ET4[+] HCs appeared at 48 hr post LY+neo, suggesting that HC regeneration happened after the recovery of sox2[+] progenitors.

In addition to sox2 antibody staining, we also used *sox2-2a-GFP* knock-in reporter (*sox2:GFP*) (*Shin et al., 2014*) to observe the recovery of sox2[+] progenitors. GFP-positive cells in neuromast, which were mostly co-labeled by sox2 antibody, were significantly decreased in LY, and recovered to normal level at 48 hr post LY+neo. In contrast, the exhausted sox2[+] progenitors were not recovered without severe injury (*Figure 1C–D*). These results indicate that sox2[+] progenitors in the zebrafish lateral line have high potential to regenerate themselves when exhausted by severe injury.

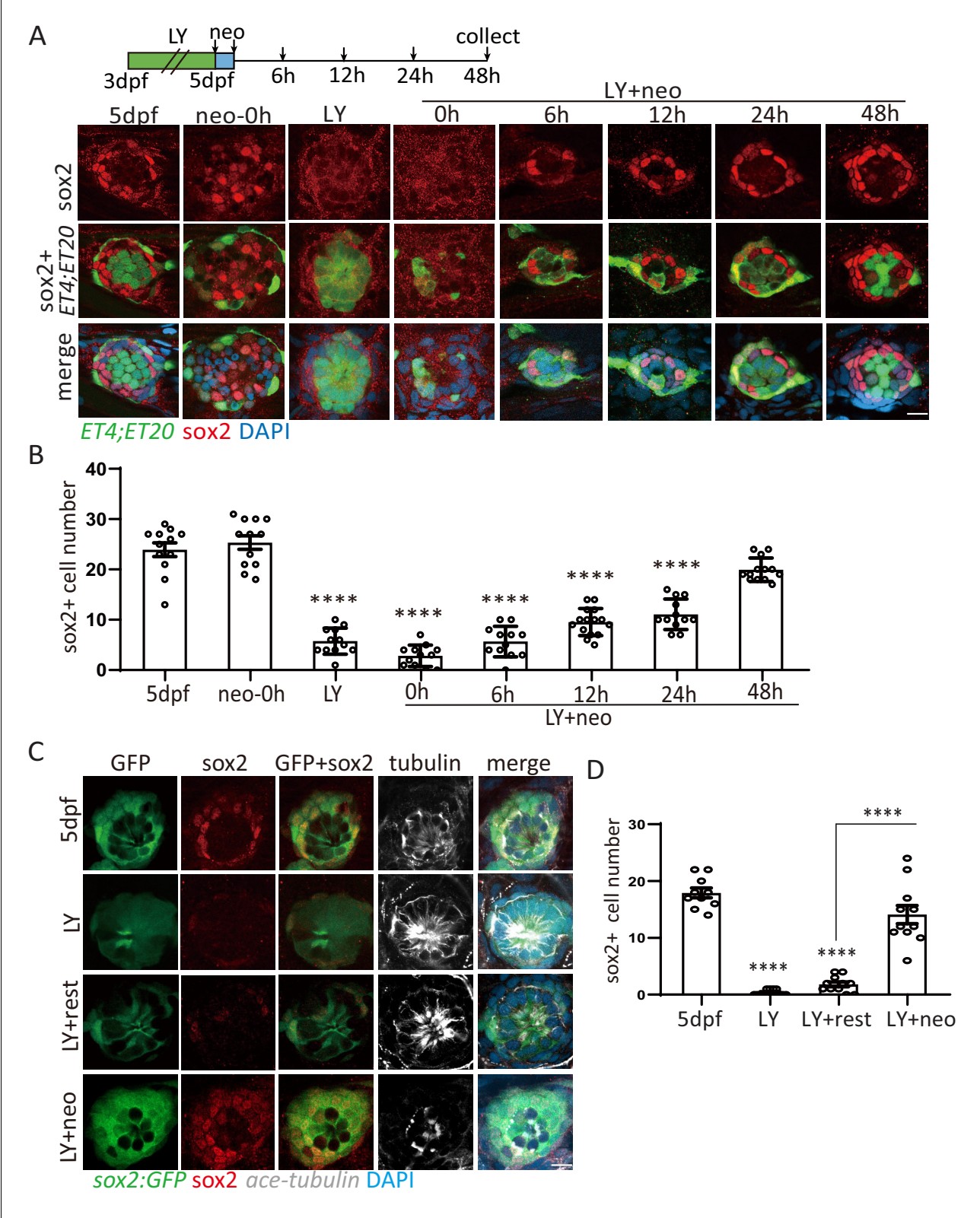

**Figure 1.** Exhausted sox2+ progenitors were able to restore quickly post severe injury. (A, B) *ET4;ET20* larvae were treated with neomycin, LY411575 (3dpf-5dpf), or neomycin following LY (LY+neo), and collected at indicated time points post neomycin treatment for sox2 immunostaining. The number of sox2+ progenitors was not affected post neo, while it was significantly decreased in LY and LY+neo-0h. The sox2+ progenitors were regenerated post LY+neo and recovered to normal level at 48 hr post LY+neo. (C, D) The *sox2:GFP* reporter was treated with LY from 3dpf to 5dpf to exhaust GFP+

*Figure 1 continued on next page*

*Figure 1 continued*

progenitors. GFP+ progenitors cannot be regenerated when resting in normal medium for 2 days post LY treatment (LY+rest). In contrast, sox2+ progenitors were quickly recovered to normal level at 2-day post LY+neo. Scale bar equals 10 μm. All groups are compared with 5dpf unless indicated. The online version of this article includes the following figure supplement(s) for figure 1:

**Figure supplement 1.** Severe injury causes damage to HCs, SCs and MCs.
**Figure supplement 2.** More proliferative SCs and MCs were induced post severe injury compared with normal injury.

During regeneration, three types of cell divisions can be detected by combining EdU staining with GFP expression in *ET4;ET20* (*Romero-Carvajal et al., 2015*). First, EdU is incorporated in differentiating cells when one HC precursor divides into two HCs (*ET4*$^+$EdU$^+$, or HC$^+$). The second type is SC proliferation in which one SC divides into two SCs (ET4$^-$ET20$^-$EdU$^+$, or SC$^+$). The third type is mantle cell (MC) proliferation in which one MC divides into two MCs (*ET20*$^+$EdU$^+$, or MC$^+$). Our results showed that proliferation in SCs or MCs post severe injury is highly increased compared with neo-induced normal injury, while HC differentiation is decreased (*Figure 1—figure supplement 2*). By time lapse, we recorded 12 times of MC or SC divisions in one neuromast from 24 hr to 40 hr post LY+neo, but none of them was differentiation (*Video 1*). These results indicate that intensive proliferation is necessary to accomplish regeneration post severe injury.

## Activated yap upregulated *lin28a* expression in *atoh1a*$^+$ HC precursors upon severe injury

We next investigated the mechanism involved in progenitor recovery. Previously, we have collected samples of regenerating neuromasts for RNA-Seq analysis (*Jiang et al., 2014*) and identified that *lin28a* was transiently upregulated post neomycin treatment (*Figure 2—figure supplement 1A*). By in situ hybridization, we verified that *lin28a* was not expressed in the developing lateral line primordium or neuromast (*Figure 2A*). Neomycin treatment and other types of HC injuries, including heavy metal (copper sulfate) or chemotherapeutic drug (cisplatin), induced sporadic *lin28a* expression (*Figure 2—figure supplement 1B*). Interestingly, LY+neo induced much higher expression of *lin28a* compared with neo alone (*Figure 2B*). Since LY+neo induced more cell death than neo as illustrated by ablation of *s100t* expressing HCs, we tested whether *lin28a* induction is proportional to the injury size. We used laser ablation to manipulate the number of injured cells in LY-treated neuromast. Low level of *lin28a* was induced when five HCs were ablated while higher level of *lin28a* was observed when sixteen HCs were ablated. Furthermore, we found that the ablation of SCs could also induce *lin28a* expression (*Figure 2—figure supplement 1C–D*).

*Atoh1a* is a master gene for HC specification and labels mostly HC precursors including differentiating HCs and young HCs (*Cai and Groves, 2015*; *Lush et al., 2019*). We noticed that *atoh1a*$^+$HC precursors were the main population that survived post LY+neo (*Figure 2C*). Using double fluorescent in situ, it was verified that *lin28a* was induced in *atoh1a*$^+$ HC precursors in both neo- and LY+neo-treated neuromasts (*Figure 2D*). We further assessed the effect of *atoh1a* expression on *lin28a* induction. We used *hs:atoh1a* to induce *atoh1a* expression after heat shock and found that neo-induced *lin28a* was increased (*Figure 2E*). In contrast, *lin28a* induction post LY+neo was completely blocked when *atoh1a* expression was inhibited in *hs:notch1a* (*Figure 2F*). All these results indicate that *lin28a* was induced in *atoh1a*$^+$ HC precursors post injury.

We further analyzed the upstream pathway that induce lin28a post injury. It's documented that Wnt activation regulates *lin28a* expression

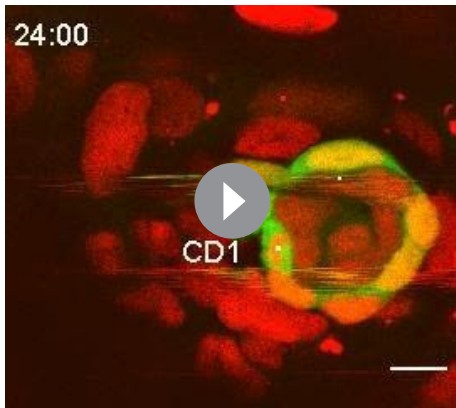

**Video 1.** *ET4;ET20;cldnB:H2Amcherry* larvae treated with LY+neo were processed for time lapse. Results showed the intensive cell divisions (CDs) during severe-injury-induced regeneration. Scale bar equals 10 μm.
https://elifesciences.org/articles/55771#video1

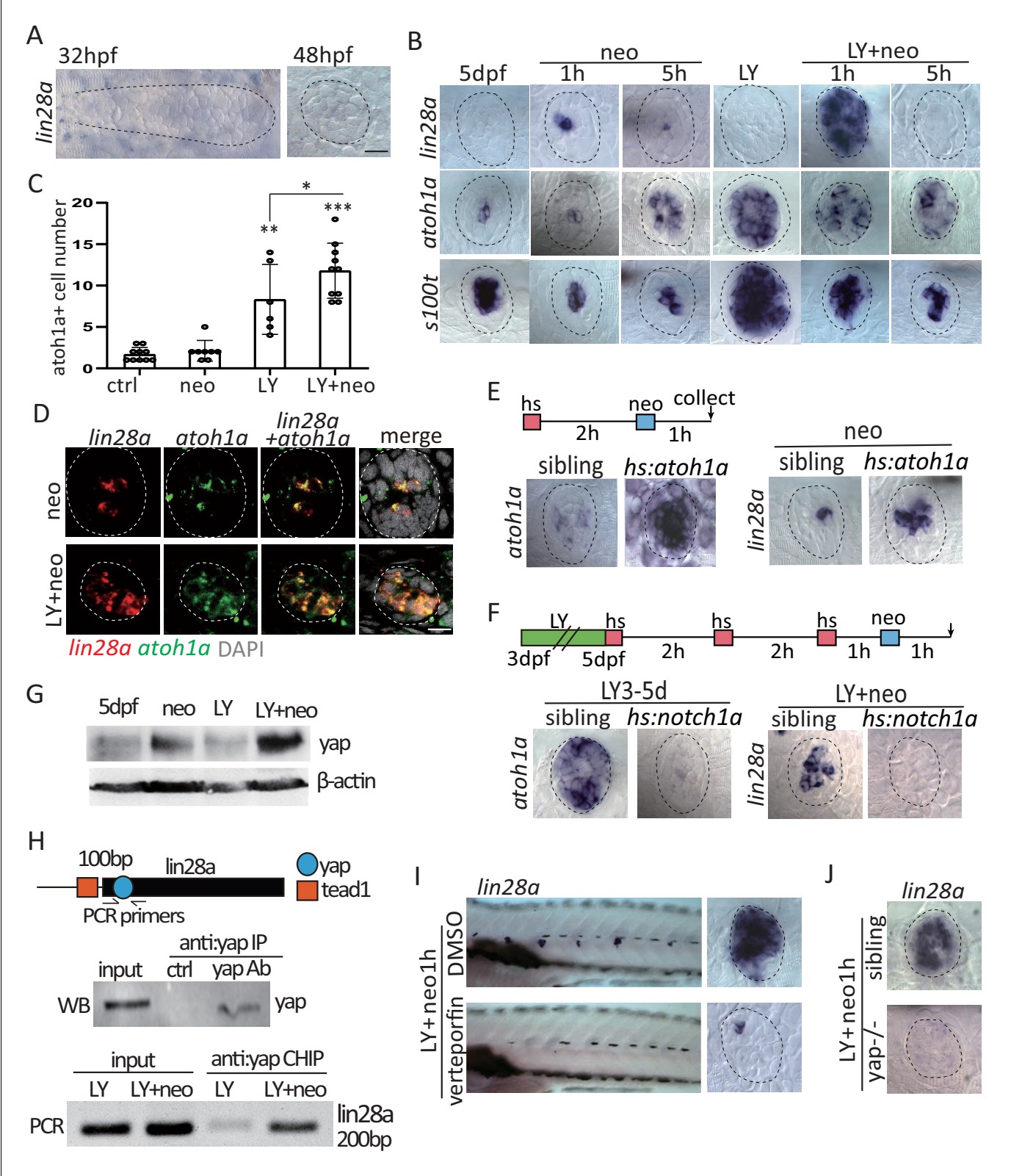

**Figure 2.** Activated y*ap* upregulate *lin28a* transcription in *atoh1a*+ HC precursors post severe injury. (**A**) *Lin28a* was not expressed in the developing lateral line primordium or neuromast. (**B**) Treatment with Notch inhibitor LY411575 from 3dpf to 5dpf increased expression *atoh1a* and *s100t*. More *lin28a* expression was observed in LY+neo-1h compared with neo-1h. No *lin28a* was detected at 5 hr post LY+neo or neo. (**C**) The number of *atoh1a*-transcribed cells detected by in situ were higher at 1 hr post LY+neo compared with LY. (**D**) Double fluorescent in situ showed that *lin28a* was co-

*Figure 2 continued on next page*

Figure 2 continued

expressed with *atoh1a* post neo or LY+neo. (**E**) Induction of *lin28a* post injury was increased when *atoh1a* was overexpressed in *hs:atoh1a*. (**F**) *Lin28a* expression was completely blocked when *atoh1a* was inhibited in *hs:notch1a*. (**G**) Western blot results showed that LY+neo induced higher *yap* expression compared with neo alone. (**H**) Motifs of *yap* and *tead1* (co-transcriptional factor of *yap*) binding sites were predicted near *lin28a* transcriptional start site. CHIP-PCR results verified that *yap* directly binds the predicted motif. (**I and J**) Inhibition of *yap* using verteporfin or *yap* mutant blocked *lin28a* induction post LY+neo. Scale bar equals 10 μm.

The online version of this article includes the following figure supplement(s) for figure 2:

**Figure supplement 1.** *lin28a* was induced by various kinds of injury and *lin28a* expression level was proportional to injury size.

**Figure supplement 2.** *Yap* was activated immediately post LY+neo.

(**Yao et al., 2016**), so we first examined whether Wnt activation acts upstream to induce *lin28a* expression post injury. We used *hs:dkk1* to inhibit Wnt activation and found that *lin28a* induction post injury was not affected (**Figure 2—figure supplement 2A**). Because Hippo pathway critically regulates regeneration of many tissues (**Gregorieff et al., 2015**; **Gregorieff and Wrana, 2017**; **Moya and Halder, 2016**), we therefore tested the expression of *yap* and *taz*, two cardinal mediators of Hippo pathway. We found the protein levels of yap was up-regulated at 1 hr post LY+neo while taz was increased until 5 hr (**Figure 2G** and **Figure 2—figure supplement 2B**). By immunostaining, more yap was also dectected in LY+neo-treated neuromast cells compared with neo. Yap expression was inhibited when *atoh1a* is inhibited with *hs:notch1a*, which suggests that yap is activated in *atoh1a*[+] HC precursors (**Figure 2—figure supplement 2D–G**). Expressions of the classic yap target genes, such as cyr61 and ctgfa, and the Hippo pathway genes, such as yap and mst2, were dramatically increased post LY+neo (**Figure 2—figure supplement 2H–I**), which was blocked with verteporfin (an inhibitor of yap-mediated transcription by blocking yap-tead1 interaction).

Interesting we found a conserved yap binding motif located at 100 bp downstream of lin28a transcription start site, where a tead1 binding motif is located nearby. We used CHIP-PCR to verify that yap directly binds to this region of *lin28a* promoter post severe injury (**Figure 2H**). In addition, LY+neo-induced *lin28a* expression was blocked in verteporfin or in yap mutant (**Figure 2I–J**). Taken together, we found that *yap* is highly activated by severe injury and directly binds *lin28a* promoter to initiate its transcription in *atoh1a*[+] HC precursors.

## Yap-lin28a pathway is necessary and sufficient to promote progenitor recovery

To determine whether *lin28a* is required for HC regeneration, we generated a *lin28a* mutant allele (*lin28åpsi37*, referred to as *lin28a-*) that harbors a deletion of five nucleotides and causes a premature stop codon within the cold shock domain (**Figure 3—figure supplement 1**). We found that *lin28a* deficiency had no effect on HC or SC number in homeostasis, nor did it affect exhaustion of sox2[+] progenitors post LY or LY+neo (**Figure 3—figure supplement 2**). Since LY+neo treatment substantially enhanced *lin28a* expression, we examined whether *lin28a* deficiency affected LY+neo-induced regeneration. We found that the regenerated HC number was decreased by *lin28a* deficiency (**Figure 3A**). In addition, the number of regenerated SCs (**Figure 3A**) and the proliferative SCs (**Figure 3B**) were both significantly decreased. The EdU-positive SC cells that were located in each quadrant with no polarization in sibling post LY+neo were almost cleared in *lin28a* mutant (**Figure 3C–D**). In contrast, the EdU-positive SC number post neo is not affected by *lin28a* deficiency (**Figure 3E**), indicating that *lin28a* is not required for neo-induced regeneration.

We also observed that yap deficiency induced similar phenotype with *lin28a* mutant. Yap mutation or inhibition with verteporfin caused proliferative deficiency post LY+neo, but has no effect on SC proliferation post neo (**Figure 3F–G** and data not shown). We have generated a transgenic line in which heat shock promoter is used to drive *lin28a* expression. Proliferative deficiency in verterporfin was rescued by *hs:lin28a* (**Figure 3G**), indicating that *lin28a* acts downstream of yap to promote SC proliferation. We next examined whether *lin28a* is required for recovery of sox2[+] progenitors post LY+neo. The number of proliferative and regenerated progenitors at 12 hr post injury were significantly reduced in *lin28a* mutant (**Figure 3H–I**).

We found that the HC number was not changed in *hs:lin28a*, indicating that *lin28a* has no effect on HC development (**Figure 4—figure supplement 1**). We next examined whether *lin28a*

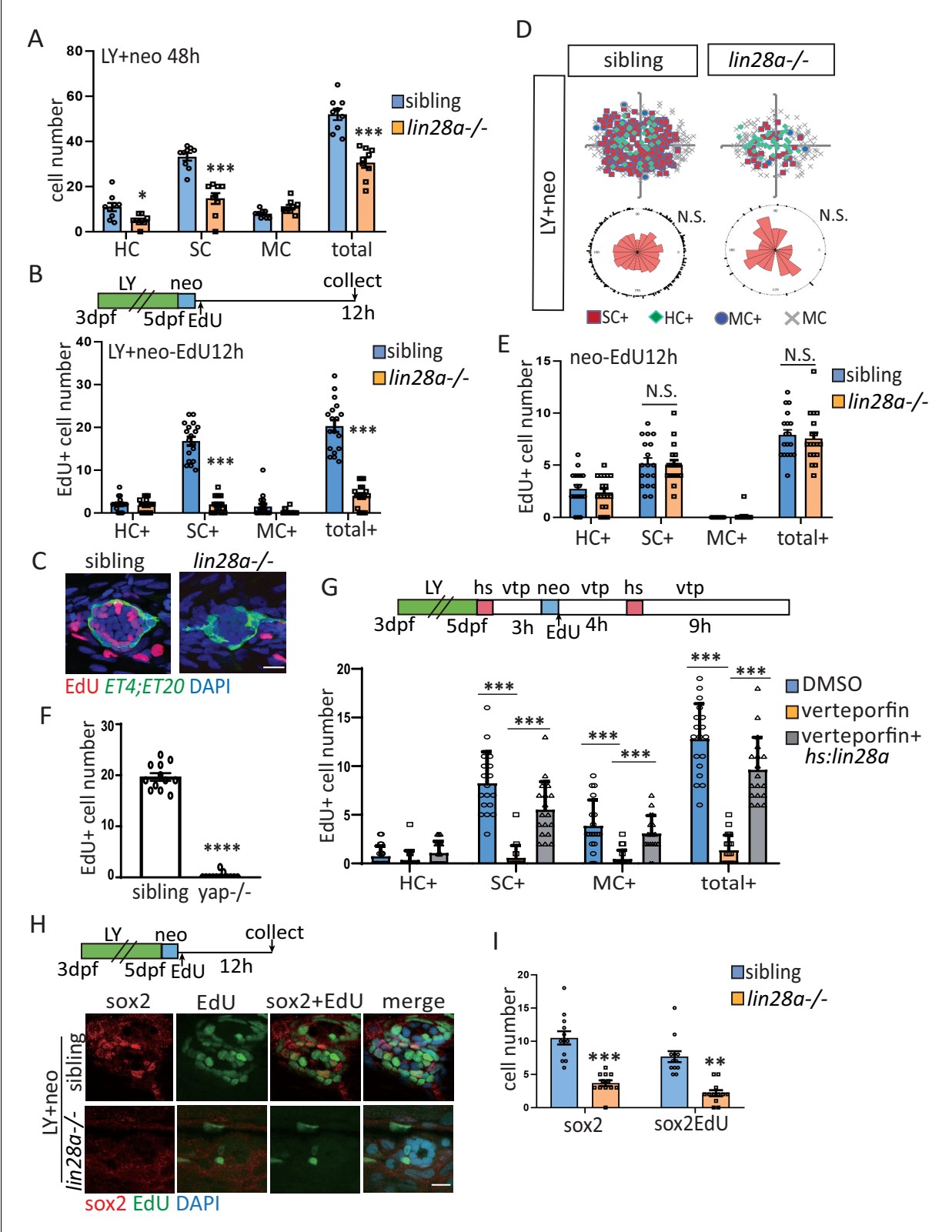

**Figure 3.** Yap-lin28a pathway is essentially required for progenitor recovery post severe injury. (**A**) The number of SCs and total cells were counted at 48 hr post LY+neo and were significantly decreased in *lin28a* mutant compared with sibling. (**B–E**) The *ET4;ET20* larvae were incorporated with EdU post neo or LY+neo treatment. Three populations labeled with *ET4*⁺EdU⁺ (HC⁺), *ET20*⁺EdU⁺ (MC⁺) or ET4⁻ET20⁻EdU⁺ (SC⁺) were counted and recorded with location information. The proliferative SCs post LY+neo were significantly decreased in *lin28a* mutant compared with sibling (**B and C**),

*Figure 3 continued on next page*

*Figure 3 continued*

while not changed in neo-induced regeneration (**E**). (**D**) EdU plots show the positions of EdU+ nuclei of 18 neuromasts superimposed on the same plane, and rose diagrams document the angular positions of SC+. The results show that the proliferative SCs are evenly distributed in each quadrant with no polarization post LY+neo. (**F**) LY+neo-induced Edu incorporation was significantly reduced in yap mutant. (**G**) The *hs:lin28a* larvae were heat-shocked and pre-treated with 10 µM verteporfin before adding neomycin. Samples were collected at 14 hr post LY+neo for EdU+ cell counting. Proliferation (SC+) is significantly decreased post LY+neo in verteporfin, which could be rescued by overexpression of *lin28a* with *hs:lin28a*. (**H, I**) The numbers of regenerated sox2+ progenitors (sox2) and proliferative progenitors (sox2EdU) post LY+neo were both reced in *lin28a* mutant. Scale bar equals 10 µm.

The online version of this article includes the following figure supplement(s) for figure 3:

**Figure supplement 1.** Create *lin28a* mutant using CRISPR.
**Figure supplement 2.** HCs and sox2+ progenitors were not affected by *lin28a* deficiency in homeostasis, LY and LY+neo.

overexpression is sufficient to promote progenitor proliferation. The number of proliferative SCs was significantly increased when *lin28a* is overexpressed in both homeostatic and neo-treated neuro-mast, whereas HC differentiation was not changed (**Figure 4A–B**). The proliferative SCs are localized in dorsal and ventral poles of neuromast in homeostasis and neo-induced regeneration (**Ma et al., 2008**; **Romero-Carvajal et al., 2015**; **Wibowo et al., 2011**). The location of proliferative SCs still remain polarized in *hs:lin28a* (**Figure 4C–D**). We further examined whether *lin28a* overexpression is sufficient to induce sox2+ progenitors, and observed higher number of proliferative progenitors in *hs:lin28a* post LY (**Figure 4E–F**).

## Atoh1a+ HC precursors gained sox2 expression through Yap-lin28a pathway

Our data indicate that *lin28a* is induced in *atoh1a*+ HC precursors post injury, but *lin28a* functions to recover sox2+ progenitors during regeneration. To address this paradox, we hypothesized that HC precursors gained sox2 expression to initiate regeneration post severe injury. We co-labeled the expressions of *atoh1a* mRNA and sox2 protein and found that the number of *atoh1a*+sox2+ cells in LY+neo was significantly increased compared with neo (**Figure 5A–B**). In addition, we used *atoh1a:TdTomato* reporter to test whether *atoh1a*+ cells expressed sox2 post severe injury. The *atoh1a:TdTomato* reporter labeled partially *ET4*-positive HCs (Figue 5C, ctrl) and also a few cells that are *ET4* negative and sox2 negative which are likely HC precursors (**Figure 5C**, yellow arrows in LY). Very few Tomato+ cells express sox2 in normal or LY-treated larvae. However, significantly more Tomato+ cells turned on sox2 expression at 12 hr and 48 hr post LY+neo and become Tomato+-sox2+ cells (**Figure 5D**). We further used time-lapse microscopy to trace *atoh1a:TdTomato* cells and found that the Tomato+ cells became HCs in both neo and LY+neo (blue dots in **Figure 5—figure supplement 1A–B**, **Video 2** and S3). However, it's only in LY+neo-induced severe injury that Tomato+ cells became sox2-positive cells labeled by *sox2:GFP* (white dots in **Figure 5—figure supplement 1B** and **Video 3**). We traced the Tomato+ cells for their fate becoming HCs, SCs or MCs (**Figure 5—figure supplement 1C–D**), and the results showed that significantly higher ratio of SCs were labeled by Tomato post LY+neo. More importantly many Tomato+ MCs were labeled post LY+neo, but none was detected in normal or neo-treated larvae, suggesting that *atoh1a*+sox2+ cells gained more potential to produce MCs.

We further tested whether HC precursors gained sox2 expression through Yap-lin28a pathway. The Yap inhibitor verteporfin decreased Tomato+sox2+ cell number post severe injury, which was rescued by overexpression of *lin28a* (**Figure 5E**). Significantly higher number of *atoh1a*+sox2+ cells were induced in LY-treated *hs:lin28a* (**Figure 5F–G**).

## MicroRNA let7 acts downstream of lin28a to activate wnt pathway for promoting regeneration

*Lin28a* has been described to regulate progenitor proliferation in developing inner ear by inhibiting *let7* microRNA processing (**Golden et al., 2015**). To interrogate the function of *let7* microRNA in regenerating lateral line, we have created *hs:let7* transgenic line. Similar with *lin28a* mutant, *hs:let7* showed the defect of SC proliferation post LY+neo (**Figure 6A**). We further found that overexpression of *let7* inhibited *hs:lin28a*-induced proliferation (**Figure 6B**), indicating that *let7* acts downstream of *lin28a* to promote regeneration.

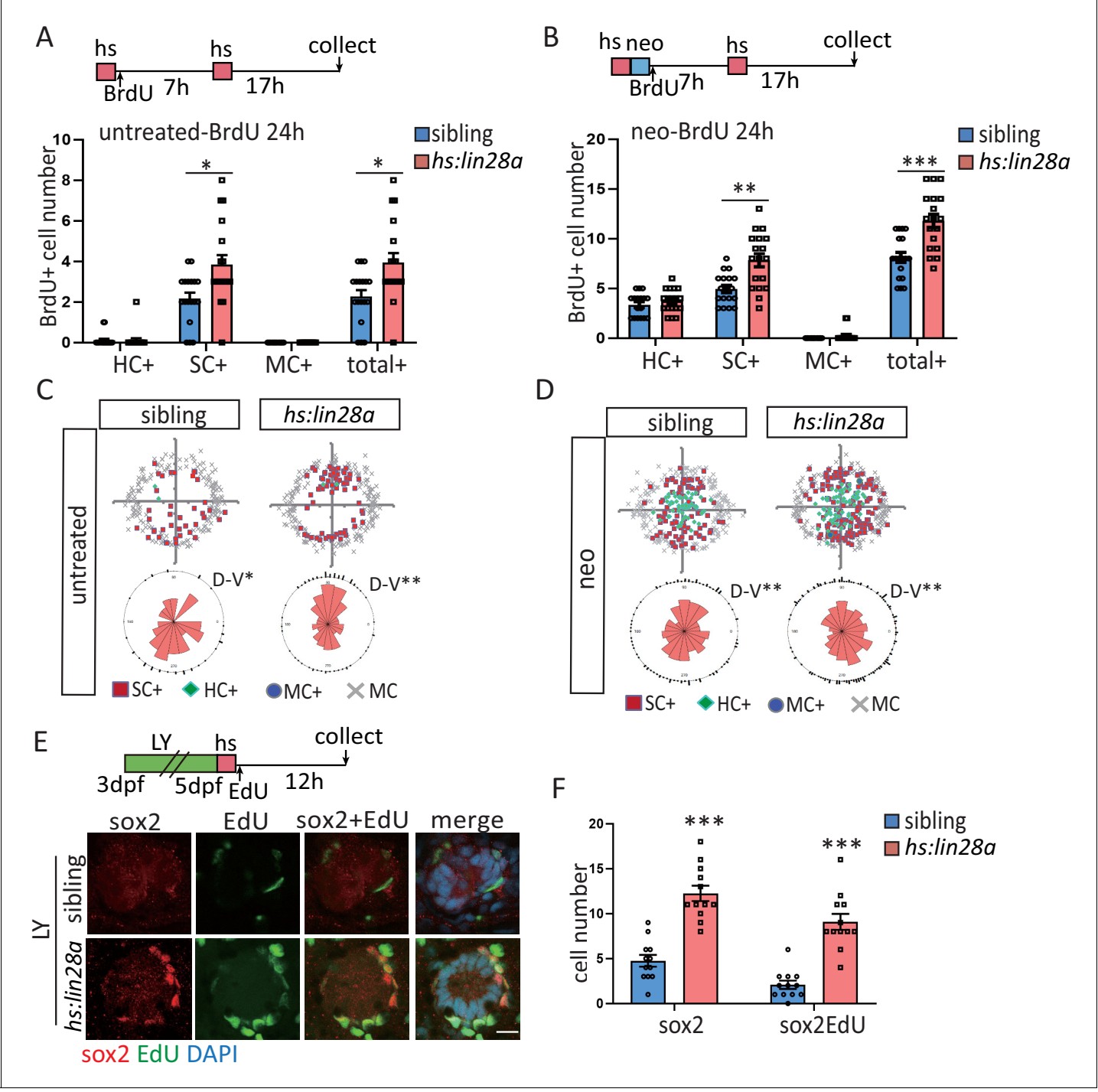

**Figure 4.** Overexpression of *lin28a* is sufficient to restore the exhausted progenitors. (A–D) *ET4;ET20;hs:lin28a* larvae were incorporated with BrdU for 24 hr following heat-shock and/or neomycin treatment. (A, B) Overexpression of *lin28a* increased number of proliferative SCs (SC+) in both untreated and neomycin conditions. (C, D) BrdU plots and rose diagrams indicate that locations of SC+ still remain dorsally and ventrally polarized in *hs:lin28a*. (E, F) OverexpressUion of *lin28a* is sufficient to partially restore the exhausted sox2+ progenitors post LY. Scale bar equals 10 µm.

The online version of this article includes the following figure supplement(s) for figure 4:

**Figure supplement 1.** Developing HCs and MCs were not affected in hs:lin28a.

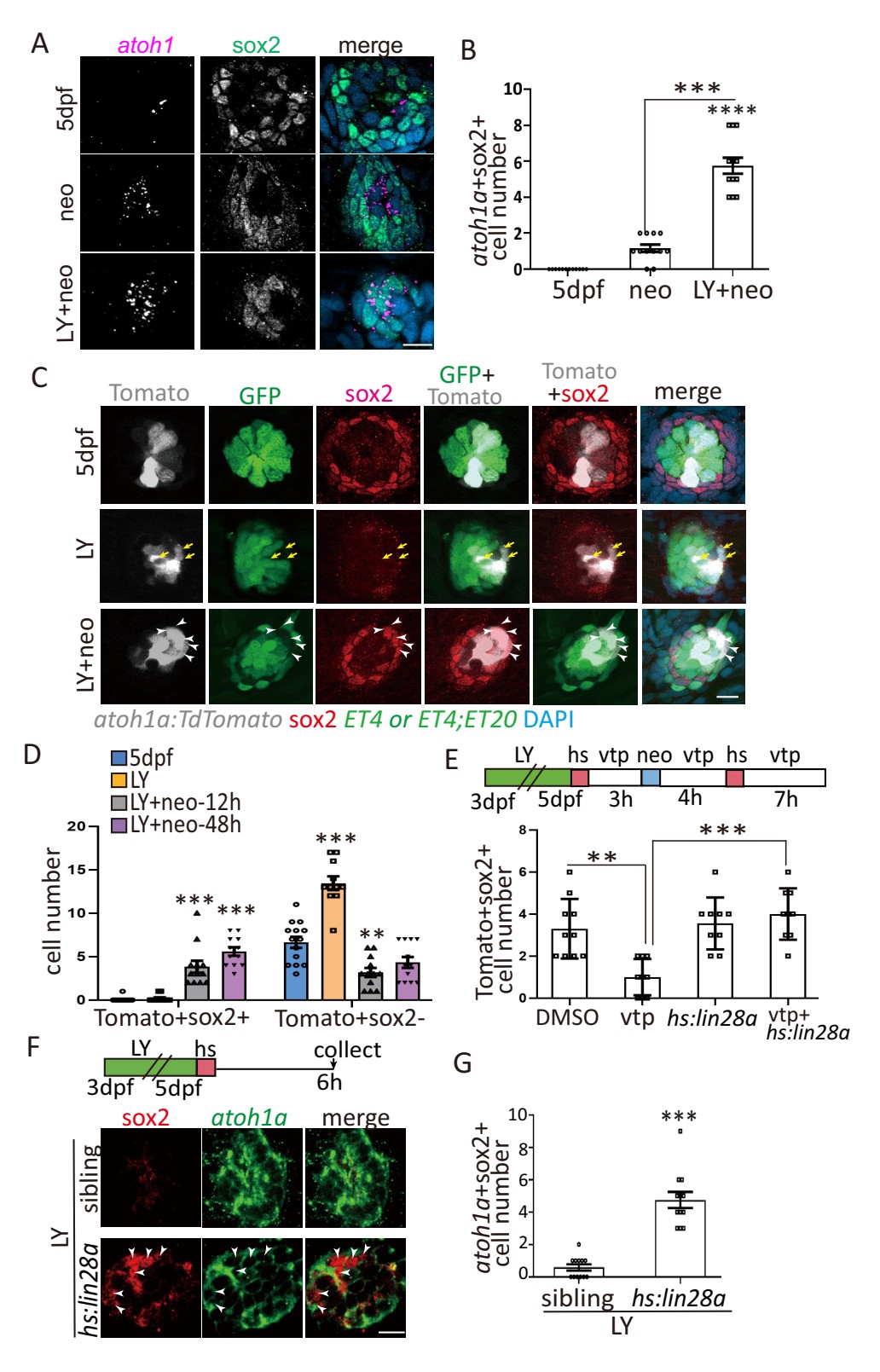

**Figure 5.** Yap-lin28a pathway promotes sox2 expression in *atoh1a*[+] HC precursors. (**A–B**) Larvae were stained with sox2 antibody and *atoh1a* RNA probe at 6 hr post neo or LY+neo treatment. The number of *atoh1a*[+]sox2[+] cells was significantly increased in LY+neo group. (**C–D**) The *atoh1a: TdTomato* larvae were used to trace *atoh1a*[+] HC precursors in ctrl, LY or LY+neo. Results showed that Tomato[+] cells labeled partial *ET4*[+] HCs and *ET4*[-]sox2[-] HC precursors (yellow arrows). However, many *atoh1a*[+] cells start to express sox2 from 12 hr post LY+neo and their numbers were

*Figure 5 continued on next page*

Figure 5 continued

significantly increased compared with normal larvae. The arrowheads in (C) pointed the *atoh1a*⁺*sox2*⁺ cells at 48 hr post LY+neo. (E) The *atoh1a: TdTomato;hs:lin28a* larvae was treated with LY+neo and verteporfin and collected for immunostaining with *sox2* antibody. The cell number of *Tomato*⁺*sox2*⁺ is decreased in verteporfin and overexpression of *lin28a* could rescue the phenotype. (F–G) The LY-treated *hs:lin28a* larvae were heat-shocked to overexpress *lin28a*. Samples were collected for staining with *sox2* antibody and *atoh1a* RNA probe. The number of *atoh1a*⁺*sox2*⁺ cells was significantly increased in *hs:lin28a* group, indicating that *lin28a* is sufficient to express *sox2* in *atoh1a*⁺ HC precursors. Scale bar equals 10 μm.

The online version of this article includes the following figure supplement(s) for figure 5:

**Figure supplement 1.** Atoh1a+ cells became SCs and MCs post severe injury.

Lin28a has no effect on Notch and Fgf pathways as *her4*, a Notch pathway gene, and *fgf3*, a Fgf pathway gene were not changed in *hs:lin28a* (*Figure 6—figure supplement 1*). We then tested whether Wnt pathway acts downstream of lin28a/let7 to promote proliferation. Overexpression of *lin28a* was sufficient to upregulate Wnt pathway genes *wnt10a* and *mycn*, which was inhibited by overexpression of *let7* (*Figure 6C*). Activation of Wnt pathway genes at 1 hr post LY+neo were blocked in *lin28a* mutant (*Figure 6D*). We further tested the function of Wnt pathway and found that *lin28a*-induced proliferation were inhibited by *hs:dkk1* in which Wnt activation is blocked (*Figure 6E*). To summarize, we found that *lin28a* activates Wnt pathway through *let7* for promoting regeneration.

## Discussion

### The zebrafish lateral line provides a valuable model for studying the mechanism underpinning progenitor regeneration

Progenitors that divide and differentiate to generate HCs in embryonic development are absent in adult mammalian inner ear, which leads to the regeneration failure post injury. In this study, we simulated the situation of progenitor absence in the zebrafish lateral line through persistent conversion of *sox2*⁺ progenitors into HCs with Notch inhibitor LY411575. By adding neomycin post LY treatment to ablate HCs, we created a severe injury model in which both HCs and progenitors were eliminated. In big contrast to mammalian inner ear, progenitors in the lateral line were able to quickly restore themselves post severe injury, with HCs being regenerated afterwards. This model provides a valuable tool to elucidate the mechanisms underpinning progenitor recovery for initiating HC regeneration.

Here, we found that yap-lin28a-let7-Wnt axis is essential to promote progenitor regeneration. *Lin28a* is not only necessary but also sufficient to induce progenitor recovery. Our findings elucidate the underlying mechanism of progenitor regeneration, and open a novel avenue of restoring progenitors to enhance mammalian HC regeneration.

In contrast to the situation of severe injury, our data showed that *yap* inhibition or *lin28a* deficiency has no effect on progenitor proliferation post neo (*Figure 3E* and data not shown). Wnt pathway, which acts downstream of *lin28a*, is required to induce progenitor proliferation. We found that Wnt pathway was highly activated immediately post severe injury, but was not activated at 1 hr post neo. Since the number of *sox2*⁺ progenitors were not reduced post neo, it seems not necessary to activate Wnt for producing more progenitors. Therefore, our results identified that Yap-lin28a pathway functions specifically in severe-injury induced regeneration for promoting progenitor recovery (*Figure 7*).

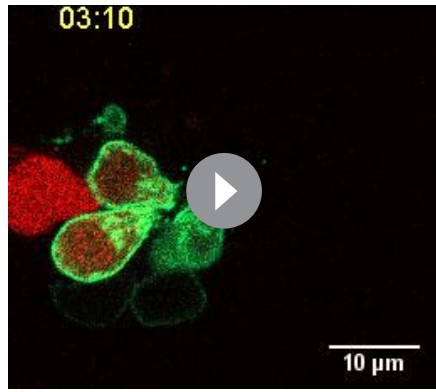

**Video 2.** The neuromast of *atoh1a:TdTomato;brn3c: GFP* larvae treated with neo was imaged for time lapse. Result showed that one Tomato⁺ cell divided and turned into two GFP⁺ hair cells post neo. Scale bar equals 10 μm.

https://elifesciences.org/articles/55771#video2

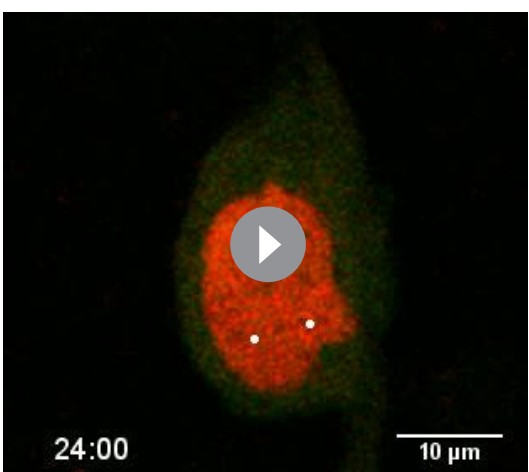

**Video 3.** The neuromast of *atoh1a:TdTomato;sox2:GFP* larvae treated with LY+neo was imaged for time lapse. One Tomato[+] cell (blue dot) divided and turned into two HCs that are GFP negative in the center. The other two Tomato[+] cells (white dots) divided and converted into four sox2[+] progenitors. Scale bar equals 10 μm. https://elifesciences.org/articles/55771#video3

## The upstream signal that activates yap post injury

Previous studies have described yap as an important regulator in mediating regeneration (*Gregorieff and Wrana, 2017*; *Moya and Halder, 2016*), but the downstream mechanism is unclear. Here, in this paper, we prove that severe injury activated *yap* directly binds *lin28a* promoter for transcriptional initiation. *Lin28a* acts downstream of *yap* to regulate progenitor regeneration.

The mechanism of *yap* activation post severe injury remains elusive. It is hypothesized that either injured cells secrete diffusible molecules leading to widespread *yap* activation, or the damage of cell junction adjacent to the injury site activates *yap* (*Moya and Halder, 2016*). We observed *lin28a* induction after laser ablation of HCs in the restricted areas. Our results showed that the *lin28a*-expressing cells were very close to the damaged HCs (*Figure 2—figure supplement 1C*), suggesting that the damage of cell junction might lead to yap-lin28a activation. In addition, we have noticed that not only HC injury but also the ablation of SCs was able to induce

*lin28a* (*Figure 2—figure supplement 1D*), indicating that the injury of either HCs or SCs triggers yap-lin28a activation. Since cell-junction-associated protein *amotl2a* has been reported restricting *yap* activity in the zebrafish lateral line primordium (*Agarwala et al., 2015*), it is possible that the loss of *amotl2a* post HC injury might result in *yap* activation. This assumption requires further investigation.

## The cellular mechanism mediated by Yap-lin28a pathway that regulates progenitor regeneration

It was reported that lin28b-let7 functions to enhance progenitor proliferation during embryonic inner ear development (*Golden et al., 2015*), but the downstream pathway is still unknown. Our data showed that lin28a-let7 functions to promote progenitor proliferation during regeneration, which is consistent with the previous finding that *lin28a* regulates retinal regeneration through *let7* (*Ramachandran et al., 2010*). Our results indicate that Wnt pathway acts downstream of lin28a-let7 for promoting progenitor regeneration. Hippo and Wnt pathway genes, such as *yap1*, *wnt10a*, *mycn*, *cyclind1*, were highly induced by Yap-lin28a post severe injury. During early development the lateral line primordium that migrates from head to tail deposits neuromasts in the trunk. Hippo and Wnt pathway genes, which are highly expressed in the leading edge of the migrating primordium containing mostly the undifferentiated mesenchymal-like progenitors (*Aman and Piotrowski, 2008*; *Kozlovskaja-Gumbrienė et al., 2017*), get silenced when the leading progenitors differentiate into SCs and HCs (*Jiang et al., 2014*; *Kozlovskaja-Gumbrienė et al., 2017*). However, the severe-injury-treated neuromast displayed the expression patterns of leading progenitors in early developing primordium, which suggests that *lin28a* reprograms the differentiated cells into the stem/progenitor cells at early developmental stage.

Although resident stem/progenitor cells are required for homeostasis and regeneration in most tissues, emerging evidence implies that differentiated cells are able to reprogram to stem/progenitor cells upon tissue damage to initiate regeneration (*Lin et al., 2018*; *Tetteh et al., 2015*). For example, atoh1[+] secretory cells in intestine are capable to dedifferentiate into lgr5[+] stem cells in irradiation-induced severe injury (*Tetteh et al., 2016*; *Tomic et al., 2018*). In colon, atoh1[+] secretory progenitors reprogram to lgr5[+] stem cells and form the entire crypts post injury while contributed minimally to other lineages in homeostasis (*Castillo-Azofeifa et al., 2019*). Since it's well-known that

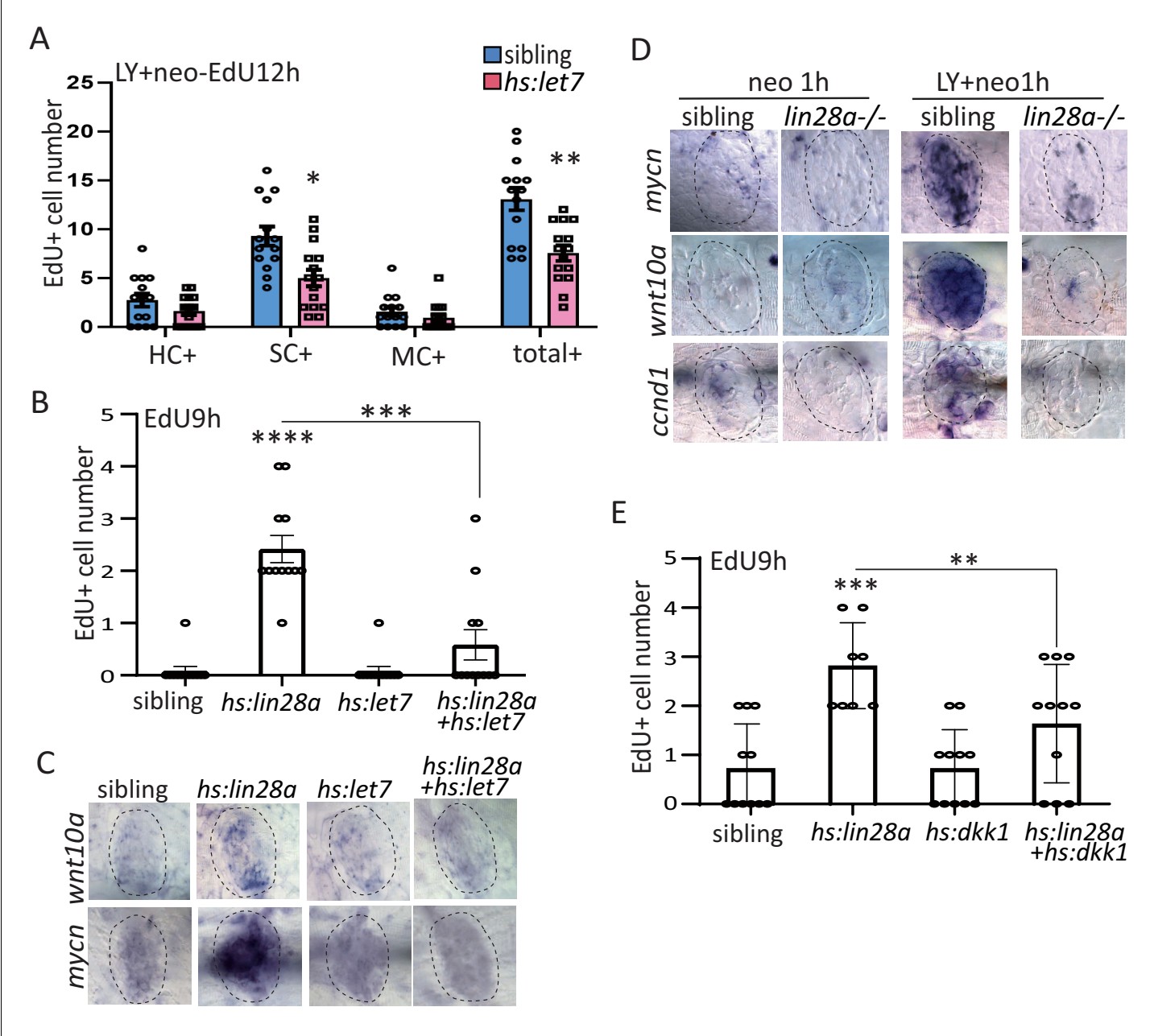

**Figure 6.** MicroRNA *let7* acts downstream of *lin28a* to activate Wnt pathway for promoting progenitor regeneration. (**A**) We created *hs:let7* transgenic line and found that it recapitulates the phenotype of *lin28a-/-* by decreasing proliferative SCs post LY+neo. (**B**) The induction of EdU+ proliferative cells was blocked by *hs:let7*, indicating that *let7* microRNA acts downstream of *lin28a* to induce proliferation. (**C**) In situ hybridization results showed that expression of Wnt pathway genes, such as *wnt10a* and *mycn*, were increased in *hs:lin28a*. The activation of Wnt pathway genes in *hs:lin28a* were blocked when *let7* was overexpressed, indicating that *let7* acts downstream of *lin28a* to inhibit Wnt pathway. (**D**) Expressions of Wnt pathway genes, such as *mycn*, *wnt10a* and *ccnd1* (*cyclind1*), were highly induced at 1 hr post LY+neo in sibling, but were not detected in *lin28a-/-*. (**E**) Inhibition of Wnt pathway with *hs:dkk1* decreased *lin28a*-induced EdU+ proliferative cells, indicating that Wnt activation acts downstream of *lin28a* to induce regeneration.

The online version of this article includes the following figure supplement(s) for figure 6:

**Figure supplement 1.** Notch and Fgf pathways were not affected by *lin28a* overexpression.

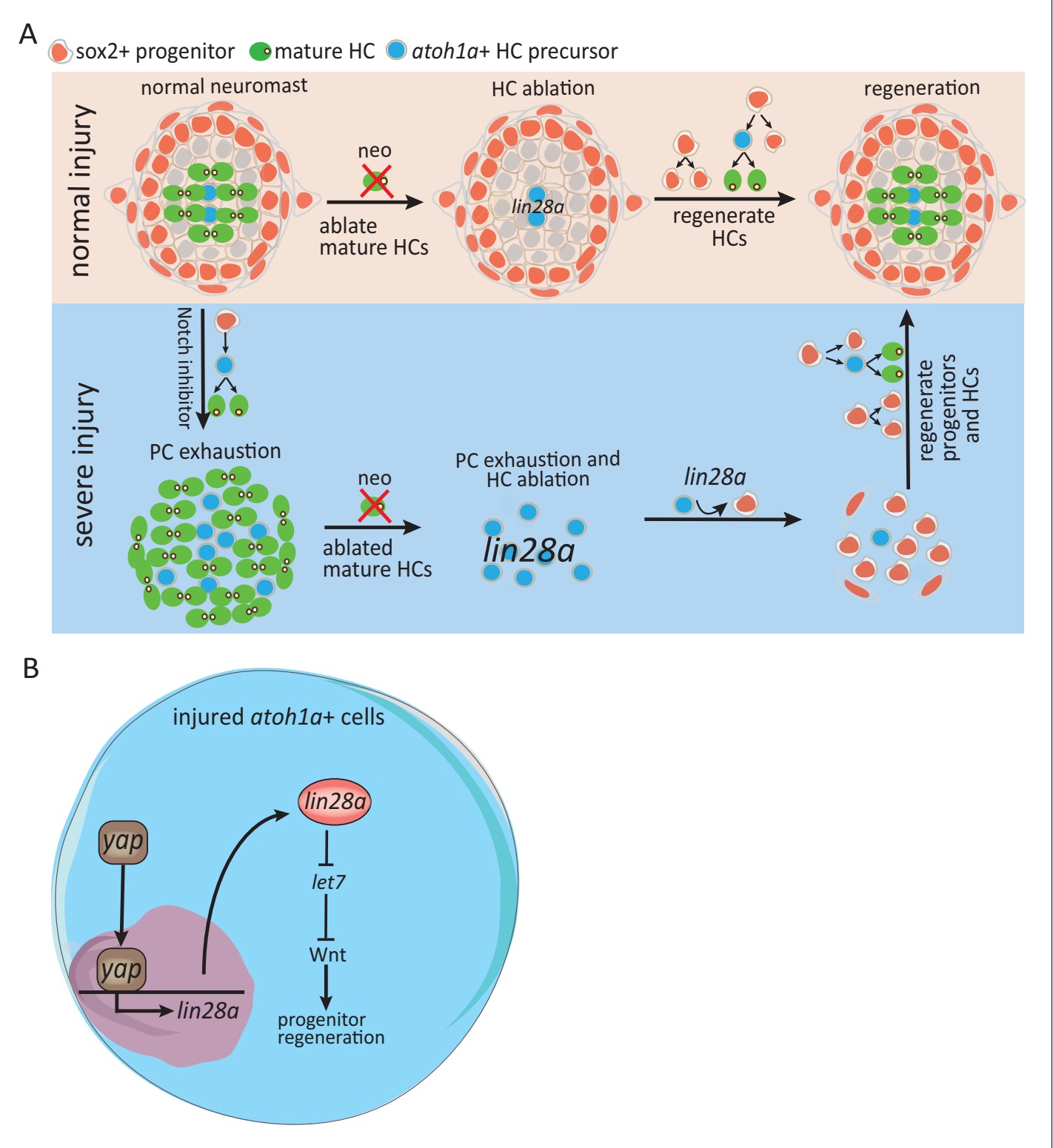

**Figure 7.** Schematic model illustrates how Yap-lin28a pathway regulates progenitor regeneration post severe injury. (**A**) It's known that sox2+ progenitors that are preserved post neo-induced HC injury divide and differentiate to regenerate HCs. To simulate the situation in mammalian inner ear where the progenitors are absent, we created a severe injury model by exhausting sox2$^+$ progenitors with Notch inhibitor and ablating mature HCs with neomycin. We found that the exhausted sox2$^+$ progenitors in severe injury have high potential to restore themselves within 48 hr, with HCs being regenerated afterwards. The atoh1a$^+$ HC precursors, the main population that survived post severe injury, were converted into proliferative progenitors

*Figure 7 continued on next page*

*Figure 7 continued*

to initiate regeneration through yap-lin28a pathway. (B) *Yap* is activated in *atoh1a*[+] HC precursors post severe injury and binds directly to *lin28a* promoter for initiating its transcription. *Lin28a* activates Wnt pathway through microRNA *let7* to promote regenerative proliferation.

lin28a, together with Oct4, Sox2 and Nanog, can reprogram somatic cells to induced pluripotent stem cells (iPSC) in mouse and human (*Yu et al., 2007*; *Moss and Tang, 2003*; *Zhang et al., 2016*), it may be involved in tissue regeneration through dedifferentiation. Our results showed that *atoh1a*[+] HC precursors were converted into sox2 expressing cells through Yap-lin28a pathway (*Figure 5*), suggesting that *lin28a* may reprogram HC precursors into sox2[+] progenitors. Further lineage-tracing analyses using cre/loxP system is necessary to verify whether HC precursors are dedifferentiated into sox2[+] progenitors post severe injury.

# Materials and methods

## Key resources table

| Reagent type (species) or resource | Designation | Source or reference | Identifiers | Additional information |
|---|---|---|---|---|
| Genetic reagent (*Danio rerio*) | *Et(krt4: EGFP)*[sqet20ET] | *Parinov et al., 2004* | sqET20; RRID:ZFIN_ZDB-ALT-070628-20 | |
| Genetic reagent (*Danio rerio*) | *Et(krt4: EGFP)*[sqet4ET] | *Parinov et al., 2004* | sgET4; RRID:ZFIN_ZDB-GENO-110323-1 | |
| Genetic reagent (*Danio rerio*) | *Tg(atoh1a: TdTomato)*[nns8] | *Wada et al., 2010* | nns8; RRID:ZFIN_ZDB-GENO-120530-1 | |
| Genetic reagent (*Danio rerio*) | *sox2-2a-sfGFP*[stl84] | *Shin et al., 2014* | stl84; RRID:ZFIN_ZDB-GENO-150721-11 | |
| Genetic reagent (*Danio rerio*) | Tg(*brn3c: GAP43-GFP*)[s356t] | *Xiao et al., 2005* | S356t; RRID:ZFIN_ZDB-GENO-200218-3 | |
| Genetic reagent (*Danio rerio*) | Tg(*hsp70l: dkk1-GFP*)[w32] | *Stoick-Cooper et al. (2007)* | W32; RRID:ZFIN_ZDB-ALT-070403-1 | |
| Genetic reagent (*Danio rerio*) | Tg(*hsp70l: myc-notch1a;cryaa: Cerulean*)[fb12] | *Zhao et al. (2014)* | fb12; RRID:ZFIN_ZDB-ALT-140522-5 | |
| Genetic reagent (*Danio rerio*) | Tg(*hsp70:atoh1a*)[x20] | *Millimaki et al. (2010)* | X20; RRID:ZFIN_ZDB-GENO-110315-10 | |
| Genetic reagent (*Danio rerio*) | *yap*[mw48] | *Miesfeld et al., 2015* | Ms48; RRID:ZFIN_ZDB-ALT-160122-5 | |
| Genetic reagent (*Danio rerio*) | Tg(*hsp70:lin28a-P2Amcherry; cmlc:GFP*)[psi30] | this paper | | Details in Fish strain information |
| Genetic reagent (*Danio rerio*) | Tg(*hsp70:let7-P2AGFP;cryaa: Venus*) | this paper | | Details in Fish strain information |
| Genetic reagent (*Danio rerio*) | *lin28a*[psi37] | this paper | | Details in Fish strain information |
| Antibody | anti-sox2 (Rabbit, polyclonal) | Abcam | Cat# Ab97959; RRID:AB_2341193 | IF(1:200) |

*Continued on next page*

*Continued*

| Reagent type (species) or resource | Designation | Source or reference | Identifiers | Additional information |
|---|---|---|---|---|
| Antibody | anti-yap (Rabbit, polyclonal) | CST | Cat# 4912, RRID:AB_2218911 | IF(1:200) |
| Antibody | anti-taz (Rabbit, polyclonal) | Abcam | Cat# Ab84927; RRID:AB_1925489 | IF (1:200), WB (1:500) |
| Antibody | anti-GFP (Mouse, monoclonal) | Invitrogen | Cat# A11120; RRID:AB_221568 | IF(1:500) |
| Antibody | Anti-yap (Mouse, monoclonal) | Santa Cruz | Cat# sc-271134; RRID:AB_10612397 | WB (1:1000) |
| Antibody | Anti-lin28a (Rabbit, polyclonal) | CST | Cat# 3978; RRID:AB_2297060 | WB (1:500) |
| Antibody | Anti-lats (Rabbit, monoclonal) | CST | Cat# 3477; RRID:AB_2133513 | WB (1:500) |
| Antibody | Anti-p-mob1 (Rabbit, monoclonal) | CST | Cat# 8699; RRID:AB_11139998 | WB (1:500) |
| Antibody | Anti-β-actin (Mouse, monoclonal) | Sigma | Cat# A1978; RRID:AB_476692 | WB (1:2000) |
| Antibody | Anti-digoxingenin POD (sheep, polyclonal) | Roche | 11207733910; RRID:AB_514500 | 1:2000 |
| Antibody | Anti-fluorescein POD(sheep, polyclonal) | Roche | 11426346910; RRID:AB_840257 | 1:2000 |
| Chemical compound, drug | EdU | Carbosynth | NE08701 | 3.3 mM |
| Chemical compound, drug | LY411575 | Santa Cruz | sc-364529 | 2 µM |
| Chemical compound, drug | Neomycin sulfate | Sigma | N6386 | 300 µM |
| Chemical compound, drug | Alexa Fluor-594 Azide | Thermo Fisher Scientific | N6386 | |
| Chemical compound, drug | Verteporfin | Selleckchem | S1786 | 5 µM |
| Chemical compound, drug | Copper sulfate | Sigma | 451657 | 50 µM |
| Chemical compound, drug | cisplatin | Sigma | 33342 | 500 µM |
| Commercial assay or kit | TSA-Cyanine 3 Reagent | PerkinElmer | SAT704A001EA | |
| Commercial assay or kit | TSA-FITC Reagent | PerkinElmer | SAT704A001EA | |
| Commercial assay or kit | dynabeads | Invitrogen | 10015D | |

## Fish strains

Tg(sqET20;sgET4) (Parinov et al., 2004), Tg(atoh1a:TdTomato)$^{nns8}$(Wada et al., 2010), sox2-2a-sfGFP$^{stl84}$ (Shin et al., 2014), Tg(brn3c:GAP43-GFP)$^{s356t}$ or brn3c:GFP(Xiao et al., 2005), Tg(hsp70l:dkk1-GFP)$^{w32}$ or hs:dkk1 (Stoick-Cooper et al., 2007), Tg(hsp70l:myc-notch1a;cryaa:Cerulean)$^{fb12}$ or hs:notch1a (Zhao et al., 2014), Tg(hsp70:atoh1a)$^{x20}$ or hs:atoh1a (Millimaki et al., 2010), yap$^{mw48}$ (Miesfeld et al., 2015) were used. To generate Tg(hsp70:lin28a-P2Amcherry;cmlc:GFP)$^{psi30}$ or hs:lin28a line, the lin28a coding sequence was cloned into the Gateway destination vector containing the hsp70 promoter. To generate Tg(hsp70:let7-P2AGFP;cryaa:Venus) or hs:let7 line, the expression cassette of pri-let-7a and pri-let-7f in UI4-GFP-SIBR backbone (Ramachandran et al., 2010) was subcloned into the Gateway destination vector containing the hsp70 promoter. To create lin28a$^{psi37}$ mutant or lin28a-/-, 50 pg Cas9 protein (PNA Bio) and 50 pg sgRNA (GAGGGTTTTCGCAGTCTGA) were injected per embryo. F0 founders were screened by genotyping F1 embryos with PCR (F: TG TTTGACATCTCTGCAGAGC, R:CACCGATCTCCTTTTGACCG) followed by Hpy188I digestion. The yap $^{mw69}$ mutant was genotyped with PCR primers (F:AGTCATGGATCCGAACCAGCACAA, R: TGCAATCGGCCTTTATTTTCCTGC) followed by HinfI digestion.

## Pharmacological inhibitors and heat-shock experiments

The γ-secretase inhibitor LY411575 (Santa Cruz sc-364529) was added to larvae at 2 µM from 3dpf to 5dpf. The inhibitor of yap-tead1 complex verteporfin (Selleckchem S1786)was added to larvae at 5 µM. Larvae at 5dpf were heat-shocked at 39°C for 30 min and sometimes heat-shock is repeated to maintain the target gene expression. hs:lin28a or hs:let7 larvae were sorted by GFP in heart or Venus in eye at 3dpf. hs:notch1a were sorted by Cerulean fluorescence in eye at 3dpf. And hs:dkk1 larvae were sorted by GFP after heat-shock. hs:atoh1a were genotyped by primers GCAGCCTGA-CAGGACTTTTC and GCTGCTCTTCCTGAAGTTGG.

### Regeneration experiments, EdU incorporation

To ablate HCs, larvae at 5dpf were treated with 300 µM neomycin (Fisher Scientific) for 30 min, 500 µM cisplatin (Sigma Aldrich, 33342) for 2 hr, or 50 µM CuSO4 (Sigma Aldrich, 451657) for 2 hr. To ablate both SCs and HCs to induce severe injury, we added 2 µM LY411575 from 3dpf to 5dpf and then treated with neomycin. Afterwards, larvae were first rinsed three times in fresh 0.5x E2 medium, and incubated in fresh medium. EdU (Carbosynth, NE08701, diluted in 3.3 mM with E2 medium containing 1% DMSO) was used to label larvae for the indicated time at 28.5°C before collecting for staining. Incorporated EdU was stained with Azide-594 (Invitrogen, N6386). The numbers and relative positions of EdU-positive cells in neuromast were analyzed as described in Romero-Carvajal et al., 2015.

### In situ hybridization

The following probes were used: atoh1a, her4.1, fgf3, wnt10a (Jiang et al., 2014), yap1, mst2 (Kozlovskaja-Gumbrienė et al., 2017), mycn, ccnd1(Yamaguchi et al., 2005), s100t (Venero Galanternik et al., 2015), dkk2 (Wada et al., 2013). lin28a was cloned with primers (F: CA TTACCATCCCGTGAAGAGGGTCCTGGTTCTG and R: CCAATTCTACCCGTGTGCAACAACACAC TCAGC) and subcloned into pPR-T4P for probe synthesis. In situ hybridization was performed as described in Jiang et al. (2014). Digoxigenin-labeled atoh1a probe and fluorescein-labeled lin28a probe were used for double fluorescent in situ. We first incubated the larvae with atoh1a probe followed by anti-digoxingenin POD antibody and colorized with TSA-FITC substrate. Then samples were incubated with lin28a probe followed by anti-fluorescein POD antibody and colorized with TSA-Cyanine 3 substrate.

### Immunostaining and live imaging

Antibodies against sox2 (Abcam, Ab97959), yap (Cell Signaling Technology, 4912), taz (Abcam, Ab84927), GFP (Invitrogen, A11120) were used for immunostaining as described in Kozlovskaja-Gumbrienė et al. (2017). Images were acquired on a Zeiss LSM780 or LSM800 confocal microscope using an Apochromat 40 × 1.1 NA objective. For time-lapse imaging larvae at 5dpf were anesthetized with tricaine and mounted in 1.2% low melting point agarose on glass bottom dishes.

## Western blot and CHIP experiments

About 30 larvae at 5dpf were lysed with 150 μl SDS buffer (63 mM Tris-HCl PH6.8, 10% glycerol, 100 mM DTT, 3.5% SDS) to extract protein for western blot. The primary antibodies used are yap (Santa Cruz, sc-271134), lin28a (CST, 3978), taz (Abcam, Ab84927), lats1 (CST, 3477), p-mob1 (CST 8699), β-actin (Sigma, A1978). CHIP experiments were performed as described in zfin (https://wiki.zfin.org/display/prot/Chromatin+Immunoprecipitation+%28ChIP%29+Protocol+using+Dynabeads). Antibody against yap (Santa Cruz, sc-271134) and dynabeads (Invitrogen, 10015D) were used to immunoprecipitate yap-bound nuclear DNA. PCR was performed with primers F:GATAATGATTGCATCACGTGAC and R:CATGCAGGATTCTTGGATGC to detect the region surrounding transcription start site of *lin28a*.

## Laser ablation

Larvae at 5dpf were mounted in agarose and numbered individually for laser ablation. Laser ablation was performed with a Chameleon Ultra II laser tuned to 800 nm. Regions of 2 μm diameter were bleached for 30 cycles in ZEN. Illumination power was adjusted as necessary to ensure that destruction of the targeted regions occurred but limited to only targeted areas. This could range anywhere from 40 mW to 400 mW and varied from sample to sample. Imaging was performed with an LD C-Apochromat 40 × 1.1 NA objective, with 0.5 μm pixel spacing and 1.6μs dwell time with 512 × 512 pixels. The numbered larvae were then recovered and fixed in 4% PFA for testing in situ individually.

## Yap expression analysis

Individual nuclei, labeled with DAPI, were automatically identified by finding local maxima on a Lorentzian of Gaussian filtered image. The area around identified points was automatically quantified in the yap channel. A yap-positive and yap-negative cell were manually selected from the vicinity of each neuromast, and only cells having intensity above the positive were counted as yap-nuclear-positive. All processings were done in ImageJ, and customized macros and plugins can be found at https://github.com/jouyun/smc-macros/blob/master/2DSpotFinder.ijm.

## Cell counting and data analysis

GFP in *ET20* was used to label MC while GFP in *ET4* was used to label HC. The SC number was counted by DAPI stained cells with no *ET4;ET20* expression. About 3–4 neuromasts from 4 to 6 fish were used for cell counting. The polarization analysis was performed as in *Romero-Carvajal et al., 2015*, and the enrichment in dorsal-ventral poles was calculated by Binomial analysis. All data are presented as the mean ± s.e.m. Statistical analyses were performed using Student's t-test for experiments with two groups. One-way ANOVA was used for more than two groups. * indicates $p < 0.05$, **$p < 0.01$, ***$p < 0.001$, ****$p < 0.0001$.

## Acknowledgements

We thank T Piotrowski and M Lush in Stowers Institute for Medical Research, B Link in Medical College of Wisconsin and L Solnica-Krezel in Washington University for providing fish lines; Sean Mckinney in Stowers Institute for support of imaging and data analysis. This work was supported by grants from National Key R and D Program of China (2018YFA0108304), National Science Foundation of China (81800164, 31871467), Guangdong Science and Technology (2018A030313497), Shenzhen Foundation of Science and Technology (JCYJ20170818103626421), the Key Research and Development Program of Guangdong Province (2019B020234002),Fundamental Research Funds for the Central Universities (19ykpy98).

## Additional information

### Funding

| Funder | Grant reference number | Author |
|---|---|---|
| Ministry of Science and Technology of the People's Republic of China | National Key R&D Program of China 2018YFA0108304 | Linjia Jiang |
| National Science Foundation | Youth Project 81800164 | Linjia Jiang |
| National Science Foundation | General Project 31871467 | Linjia Jiang |
| Guangdong Science and Technology Department | basic research project 2018A030313497 | Linjia Jiang |
| Guangdong Science and Technology Department | The key Research and Development Program of Guangdong Province 2019B020234002 | Meng Zhao |
| Shenzhen Foundation of Science and Technology | JCYJ20170818103626421 | Meng Zhao |
| Fundamental Research Funds for the Central Universities | 19ykpy98 | Linjia Jiang |

The funders had no role in study design, data collection and interpretation, or the decision to submit the work for publication.

### Author contributions

Zhian Ye, Zhongwu Su, Data curation, Formal analysis, Methodology; Siyu Xie, Xi Xu, Data curation, Formal analysis; Yuye Liu, Data curation; Yongqiang Wang, Data curation, Methodology; Yiqing Zheng, Meng Zhao, Supervision, Writing - review and editing; Linjia Jiang, Conceptualization, Supervision, Funding acquisition, Writing - review and editing

### Author ORCIDs

Linjia Jiang (iD) https://orcid.org/0000-0001-8854-2610

### Decision letter and Author response

Decision letter https://doi.org/10.7554/eLife.55771.sa1
Author response https://doi.org/10.7554/eLife.55771.sa2

## Additional files

### Supplementary files

• Transparent reporting form

### Data availability

All data generated or analysed during this study are included in the manuscript and supporting files.

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
