## [Decision Letter]

**Acceptance summary:**

This paper examines regeneration following hair cell damage in the developing zebrafish lateral line, a tractable model system for hearing loss. The authors use genetic and pharmacological approaches to demonstrate a role for a key signalling pathway in the recovery of lateral line hair cells following chemically-induced injury.

**Decision letter after peer review:**

[Editors’ note: the authors submitted for reconsideration following the decision after peer review. What follows is the decision letter after the first round of review.]

Thank you for submitting your work entitled "Yap-lin28a axis reprograms committed hair cells to progenitors for initiating regeneration" for consideration by *eLife*. Your article has been reviewed by three peer reviewers, and the evaluation has been overseen by a Reviewing Editor and a Senior Editor. The following individuals involved in review of your submission have agreed to reveal their identity: Shawn Burgess (Reviewer #3).

Our decision has been reached after consultation between the reviewers. Based on these discussions and the individual reviews below, we regret to inform you that your work will not be considered further for publication in *eLife*.

The manuscript tackles the interesting topic of hair cell regeneration in the lateral line of the zebrafish, using an injury model to examine the role of a yap-lin28a pathway in regenerating cells. The reviewers have pointed out several strengths of the study, but also raise a number of concerns. It was felt that there would be significant additional work necessary to meet these concerns, and that some of the conclusions were overstated. The reviewers have a number of specific suggestions, and we hope these will be useful to you in moving the work forward. The full reviews are appended below.

Reviewer #1:

This study uses a novel injury model to deplete both hair cells and surrounding supporting cells in the larval zebrafish lateral line, allowing the researchers to more closely mimic the mature mammalian inner ear, which does not contain viable progenitor cells. The experiments overall are sound, and the team makes elegant use of several heat shock lines to temporally regulate gene expression during damage and regeneration. The primary findings are that the lateral line is capable of regeneration after a severe injury, and that this regeneration is dependent on the yap-lin28-let7-Wnt pathway, while this pathway is not required for regeneration if supporting cells are still intact. This finding is novel and exciting, as the authors have identified a pathway that is necessary for regeneration in the absence of progenitor cells.

1) The findings are clearly supported by the data, although I have a reservation about the conclusion that "yap directly binds *lin28a* promoter for transcriptional initiation." The authors convincingly show that yap can bind the *lin28a* promoter, and that *lin28a* is sufficient to induce regeneration, but not that yap binding of the *lin28a* promoter is necessary for the regenerative response.

2) My other concern is the conclusion that sox2^+^/*atoh1a*^+^ cells present after LY+neo treatment truly represent dedifferentiated cells. The images in Figure 5E show that a brn3c^-^ cell expressing the *atoh1a* reporter dTomato can divide to produce two hair cells, even without a severe injury. These data suggest that *atoh1a* can label a subset of progenitor cells in the lateral line, even if these cells are sox2^-^. I agree that their LY+neo injury model increases the number of sox2^+^/*atoh1a*^+^ cells, but I am not fully convinced that these cells are dedifferentiated hair cells; they may be an intermediate progenitor cell or hair cell precursor. Similarly, I am not convinced that the cells expressing weak tubulin label in Figure 6 are dedifferentiated cells; these could be newly born hair cells, since EdU is retained following cell division. I do not think additional experiments are necessary, but rather suggest toning down the conclusions.

Reviewer #2:

This manuscript reports work done in zebrafish aimed at further clarifying the mechanism underlying hair-cell regeneration.

The authors report that *atoh1a*^+^ committed hair cells de-differentiate into sox2^+^ progenitors upon injury, which initiates regeneration through a yap-lin28a-let7-Wnt pathway. They claim that their findings "shed light on restoration of progenitors by dedifferentiation to initiate HC regeneration in mammals."

I do not share their persuasion, however.

Their data does not demonstrate hair-cell de-differentiation because *atoh1a*^+^ cells can be hair-cell progenitors that are constantly normally produced in neuromasts, or cells whose *atoh1a* expression is transitory and may not stabilise to eventually become progenitors or hair cells.

Therefore, I cannot support the publication of this work.

The authors claim that in a developing neuromast, *atoh1a* marks committed sensory hair cells. Yet, there is published data showing that this is not the case, as some non-sensory supporting cells express *atoh1a*.

Their sentence "powerful capacity to regenerate HCs sustains after multiple rounds of damage…" references Cruz et al., 2015. This in one of two concurrent papers showing regeneration after multiple rounds of hair-cell ablation. The second is Pinto-Teixeira et al., 2015, which should be cited as well.

They state that the above papers "indicates that a population of stem/progenitor cell pool exists to self-renew and regenerate." However, there is ample evidence that the entire supporting cell population is able to convert to a stem cell-like is necessary. In other words, a stem cell population as a persistent cellular state has not been demonstrated.

"Sox2 marks the proliferative progenitors that produce both SCs and HCs in homeostatic and regenerative NMs (21)."

This is true, but only because sox2 is expressed by nearly every non-sensory cell. Because sensory cells are postmitotic, their conclusion is obvious.

"These results indicate that sox2^+^ progenitors in neuromast have high potential to restore themselves when exhausted by severe injury."

Again, true and obvious because sox2 is expressed by nearly every non-sensory cell.

"These results indicate that intensive proliferation is necessary to facilitate regeneration post severe injury."

This has been shown by no less than 10 previous papers already.

Nuclear localized yap was inhibited when *atoh1a* is inhibited with *hs:notch1a*, which suggests that yap is activated in *atoh1a*^+^ committed HCs.

This is puzzling because hair cells are postmitotic. In addition, is nuclear localization of yap takes place in *atoh1a*^+^ cells and activated Notch blocks the production of these cells, it is only obvious that the authors will see no nuclear localized yap.

"Taken together, we found that yap is highly activated by severe injury and binds directly to *lin28a* promoter to initiate its transcription in *atoh1a*^+^ committed HCs."

I could not find direct evidence that yap binds *lin28a* promoter. It would help me if the authors pointing the data supporting the claim.

"We found that the regenerated HC number was decreased by *lin28a* deficiency (Figure 3A)."

What I see is that the entire neuromast was nearly half the size. Reduced hair-cell numbers may be an indirect consequence of lower proliferation, cellular extrusion or increased cell death.

"The proliferative SCs are localized in dorsal and ventral poles of neuromast in homeostasis and neo-induced regeneration"

They reference Romero-Carvajal et al., 2015. Yet, earlier reports of the patterns of supporting-cell proliferation are: Ma et al., 2008, and Wibowo et al., 2011.

"The *atoh1a*:*dTomato* reporter labeled a part of but not all *ET4* positive mature HCs in normal condition, indicating that the reporter is very mosaic (Figure 5C, ctrl)."

An alternative interpretation, supported by findings of Wibowo et al., 2011, is that *atoh1a*:*dTomato* is expressed by supporting cells that may not advance to the hair-cell progenitor or hair-cell fates.

"However, more than half of the dTomato^+^ cells turned on sox2 expression at 12h and 48h post LY+neo and become dTomato^+^sox2^+^ cells (Figure 5D), suggesting that *atoh1a*^+^ cells were converted to sox2^+^ progenitors upon severe injury."

Again, because sox2 is expressed by nearly every non-sensory cell, it is not surprising that some cells will co-express *atoh1a* and sox2.

"However, it's only in LY+neo-induced severe injury that dTomato^+^ cells converted to sox2^+^ cells labeled by *sox2:GFP* (white dots in Figure 5F and Video 3), which confirms the dedifferentiation of *atoh1a*^+^ cells into sox2^+^ progenitors.”

This statement is not supported by evidence presented in the manuscript.

Reviewer #3:

Ye et al. have submitted a manuscript that describes the interplay between the expression of sox2, lin28a, and *atoh1a* in the context of a zebrafish regenerating lateral line neuromast. The authors show that under normal conditions, *lin28a* plays at best a minor role in regeneration, it is induced in a small number of cells but a genetic knockout of *lin28a* does not have a measurable effect on lateral line hair cell regeneration after ablation with neomycin. In contrast, when notch signaling is inhibited, lateral inhibition from hair cells is blocked and you see supernumerary hair cells at the expense of supporting cells, which are the normal source of new hair cells after ablation. Under these conditions, the authors show atoh1 positive cells will induce *lin28a* and revert to a more support cell-like identity. This phenomenon is mediated via yap signaling and downstream of *lin28a* let7 is blocked, triggering Wnt signaling. When the support cells are depleted by notch inhibition, now a *lin28a* KO has a significant negative effect on hair cell regeneration. In general this is a very carefully done set of experiments that identifies an interesting gene pathway that would be difficult to recognize under typical conditions used to study regeneration. I have one significant additional experiment I would like to see.

1) The hypothesis and the evidence point to *lin28a* triggering down-regulation of *atoh1a* and a return to a support cell fate. It should therefore be true that in the earlier developing larvae, if you induced *lin28a* expression ectopically, you should be able to inhibit natural differentiation of hair cells. Part of the model presented is that *lin28a* dedifferentiated cells contribute to new supporting cell populations too. Is it possible that the *lin28a* cells can only go "backwards" far enough to divide but not far enough to become support cells again? Under those circumstances the new support cells could come from the mantle cells which have been shown to be capable of fully regenerating all cell types in the neuromast. If it is really complete dedifferentiation, there should be more supporting cells in these larvae and not cells that are neither HC nor SC. It would be interesting if there could be lineage tracing of "dedifferentiated" atoh1^+^ cells. Can they actually become supporting cells?

[Editors’ note: further revisions were suggested prior to acceptance, as described below.]

Thank you for submitting your article "Yap-lin28a axis targets let7-Wnt pathway to restore progenitors for initiating regeneration" for consideration by *eLife*. Your article has been reviewed by two peer reviewers, and the evaluation has been overseen by a Reviewing Editor and Kathryn Cheah as the Senior Editor. The following individuals involved in review of your submission have agreed to reveal their identity: Shawn Burgess (Reviewer #1).

The reviewers have discussed the reviews with one another and the Reviewing Editor has drafted this decision to help you prepare a revised submission. Please aim to submit the revised version within two months, but we are happy to extend this timeframe if needed, given the current situation which may restrict experimental work time.

Summary:

In the present study, Zhian Ye and colleagues took advantage of the zebrafish lateral line to further dissect the mechanisms enabling the mechanosensory organs of some vertebrates to regenerated while other, including the mammalian inner ear cannot. The zebrafish lateral line presents two major advantages for this study: (i) the mechanosensory organs can almost entirely regenerate and (ii) severe damages can be perform suing chemical inhibitor treatments. Here the authors show that:

1) Sox2 positive progenitors can largely regenerate after being fully illuminated in so-called severe injury after a combination of treatment with a Notch inhibitor (to promote the differentiation of support cells into hair cells) and an antibiotic (to damage the hair cells).

2) The regenerating sox2-positive cells derive from *atoh1a*-positive, committed hair cells.

3) The Hippo signaling pathway effector yap is upregulated upon severe injury, is required for regeneration of sox2-positive cells and induces the expression of lin28a.

4) *Lin28a* is necessary and sufficient to induce the regeneration of sox2-positive progenitors.

5) Progenitor regeneration results from a yap1-Lin28a-Let7-sox2-Wnt axis.

Essential revisions:

1) Improve the Western blots to strengthen the data here. At the moment, the immunostainings are not clear, and the increase in yap1 is not clear on the Western blot. Indicate Molecular Weight on the blot.

2) Confirm the specificity of your antibodies by checking on yap1 mutants.

3) Confirm the specificity of the verteporfin inhibitor by comparing to the phenotype of yap1 mutants.

4) Please attend to all the listed minor revisions and, where possible, to any other major revisions, such as citation of additional studies in the Introduction (or provide a rebuttal of these points).

The full reviews are appended below for your information, in particular, please refer to the detailed comments from reviewer 2.

Reviewer #2:

The study reveals an interesting, new mechanism driving neuromast regeneration upon severe injury. While I think the study is potentially interesting for *eLife* readership, I feel some parts are not as solid as others, in particular the part on the role of yap1. Please see my specific comments below.

1) The Introduction is short and miss to refer to some important studies that have been published on similar questions including, for example, work from the Lopez-Shier lab (PintoTeixeira et al., 2015, ViaderLlargues et al., 2018, for example)

2) Using *hs:notch1a* to block *Atoh1a* function is not ideal. The authors should use *atoh1a* mutants which are available or Morpholino which have been published.

3) The authors should demonstrate the specificity of the *yap1* and taz antibodies they are using by showing the absence of signal in respective mutants (available). These controls should be shown both for western-blot and immunostaining.

4) On the western-blot in Figure 2—figure supplement 2B, taz seems to be much more upregulated than yap1. Why do the authors then focus exclusively on yap1?

5) The yap signal and the difference between control and treated in Figure 4C and 4E are not convincing.

6) "The induction of yap and mst2 was blocked by verteporfin, an inhibitor of yap-mediated transcription by blocking yap-tead1 interaction": why are the expression of yap and mst2 blocked upon inhibition of yap-mediated transcription? Has it previously been shown that yap and mst2 are transcriptional targets of yap? The authors should look at classical yap/taz transcriptional targets.

7) Several yap and taz zebrafish mutants have been published. The authors should perform experiments in these mutants instead of – or in addition to – verteporfin experiments.

8) "We used CHIP-PCR to verify that yap binds to a region 100bp downstream of *lin28a* transcription start site". How did the author choose this region?

9) Figure 3A: does the Y-axis indicate EdU cell number (as indicated on the figure) or absolute numbers of HS, SC and MC (as the text suggests)?

10) "The EdU positive SC cells that were located in each quadrant with no polarization in sibling post LY+neo were almost cleared in *lin28a* mutant (Figure 3C-D)": The link to polarization all the sudden here is not clear.

11) Figure 3F: the authors should use yap mutants in addition to Verteporfin treatment.

---

## [Author Response]

[Editors’ note: the authors resubmitted a revised version of the paper for consideration. What follows is the authors’ response to the first round of review.]

The manuscript tackles the interesting topic of hair cell regeneration in the lateral line of the zebrafish, using an injury model to examine the role of a yap-lin28a pathway in regenerating cells. The reviewers have pointed out several strengths of the study, but also raise a number of concerns. It was felt that there would be significant additional work necessary to meet these concerns, and that some of the conclusions were overstated. The reviewers have a number of specific suggestions, and we hope these will be useful to you in moving the work forward. The full reviews are appended below.

In response to the reviewers, we have added several important references, copyedited the text, toned down some of our conclusions. Based on suggestions of reviewer 1, I toned down our conclusions about dedifferentiation. I changed the title to “Yap-lin28a axis targets let7-Wnt pathway to restore progenitors for initiating regeneration”. Instead of drawing the conclusion that *atoh1a*^+^ HC precursors were dedifferentiated, we now changed it to that *atoh1a*^+^ HC precursors gained sox2 expression post severe injury. I have deleted all the data with acetylated tubulin (original Figure 6D-G) since a reference (Liu et al., 2018) I have just read showed that it labels not only HCs but also SCs. Therefore, it’s not appropriate to use acetylated-tubulin as a specific marker for HC to test the fate change. Based on suggestions of reviewer 3 I have changed the nomination of *atoh1a*^+^ cells from committed HCs into HC precursors since *atoh1a* expression is shut down in mature HCs.

To answer the reviewer’s questions, we have made new experiments and integrated the results to the updated figures.

1) As pointed by all reviewers, we have used *atoh1a:dTomato* reporter to trace dTomato expression in HCs, SCs and MCs in normal condition, post normal and severe injury. Our new data in updated Figure 5—figure supplement 1C-D showed that *atoh1a* label mostly HCs and a small number of SCs in normal and neo-treated larvae. Higher number of *atoh1a*^+^ SCs was detected in LY+neo compared with neo. More importantly, many MCs were also labeled with *atoh1a* post LY+neo, which was not detected in normal or neo. This piece of data suggests that *atoh1a*^+^ cells behaved like progenitor/stem cells that give rise to many support cell types post severe injury.

2) As suggested by reviewer 3, we have tested *hs:lin28a* for HC differentiation during development. Our results showed that *lin28a* expression has no effect on developing HC.

I believe that the quality of the revised version is highly improved thanks to the reviewers’ comments.

Reviewer #1:[…]1) The findings are clearly supported by the data, although I have a reservation about the conclusion that "yap directly binds lin28a promoter for transcriptional initiation." The authors convincingly show that yap can bind the lin28a promoter, and that lin28a is sufficient to induce regeneration, but not that yap binding of the lin28a promoter is necessary for the regenerative response.

Verteporfin is a specific inhibitor that blocks the transcriptional activity of yap by inhibiting yap and tead interaction. Our data in Figure 3F showed that verteporfin inhibited regenerative proliferation which mimicked the phenotype of *lin28a* mutant. This piece of data indicated that yap binding of the *lin28a* promoter is necessary for the regenerative response. We have also tested yap mutants for regeneration post severe injury. Unfortunately, they all died after adding EdU post LY+neo, possibly because of their weakness caused by vascular and heart defect.

2) My other concern is the conclusion that sox2^+^/atoh1a^+^ cells present after LY+neo treatment truly represent dedifferentiated cells. The images in Figure 5E show that a brn3c^-^ cell expressing the atoh1a reporter dTomato can divide to produce two hair cells, even without a severe injury.

It’s known that the lateral line HCs are born in pairs from the terminal mitotic division of *atoh1a*^+^ HC precursors in development and post neo (Pinto-Teixeira et al., 2015). Based on reviewer 3’s suggestion, we have changed the nomination of *atoh1a*^+^ cells from committed HCs to HC precursors since mature HCs lose *atoh1a* RNA expression. Our data (blue dots in updated Figure 4—figure supplement 1A-B) are consistent with this conclusion by showing that one *atoh1a*^+^ precursor divided and became two brn3c^+^ HCs post neo.

These data suggest that atoh1a can label a subset of progenitor cells in the lateral line, even if these cells are sox2^-^. I agree that their LY+neo injury model increases the number of sox2^+^/atoh1a^+^ cells, but I am not fully convinced that these cells are dedifferentiated hair cells; they may be an intermediate progenitor cell or hair cell precursor.

Thank you to the reviewer for suggesting toning down the conclusions. Our new data in updated Figure 5—figure supplement 1C-D showed that *atoh1a* label mostly HCs and a small number of SCs in normal and neo-treated larvae. Higher number of *atoh1a*^+^ SCs was detected in LY+neo compared with neo. More importantly, many MCs were also labeled with *atoh1a* post LY+neo, which was not detected in normal or neo-treated conditions. This piece of data suggests that *atoh1a*^+^ cells behaved like progenitor/stem cells that give rise to many support cell types post severe injury. However, the lineage tracing experiment using cre/loxP system is necessary to confirm whether *atoh1a*^+^ cells are dedifferentiated or not.

Since we were not able to do this experiment because of technology reason, we have already toned down the conclusions in our manuscript. Now the dedifferentiation conclusion was changed to that *atoh1a*^+^ HC precursors gained sox2 expression post severe injury.

Similarly, I am not convinced that the cells expressing weak tubulin label in Figure 6 are dedifferentiated cells; these could be newly born hair cells, since EdU is retained following cell division. I do not think additional experiments are necessary, but rather suggest toning down the conclusions.

We have decided to delete all the data with acetylated tubulin since it’s not a specific marker for HC. A reference I have just read described that not only HCs but also SCs have acetylated tubulin expression in mouse inner ear (Liu et al., 2018, Eur J Histochem). Therefore, it’s not appropriate to use acetylated tubulin as a specific HC marker to test whether HCs were co-labeled with sox2 and EdU.

Reviewer #2:This manuscript reports work done in zebrafish aimed at further clarifying the mechanism underlying hair-cell regeneration.The authors report that atoh1a^+^ committed hair cells de-differentiate into sox2^+^ progenitors upon injury, which initiates regeneration through a yap-lin28a-let7-Wnt pathway. They claim that their findings "shed light on restoration of progenitors by dedifferentiation to initiate HC regeneration in mammals."I do not share their persuasion, however.Their data does not demonstrate hair-cell de-differentiation because atoh1a^+^ cells can be hair-cell progenitors that are constantly normally produced in neuromasts, or cells whose atoh1a expression is transitory and may not stabilise to eventually become progenitors or hair cells.Therefore, I cannot support the publication of this work.The authors claim that in a developing neuromast, atoh1a marks committed sensory hair cells. Yet, there is published data showing that this is not the case, as some non-sensory supporting cells express atoh1a.

I agree with the reviewer that non-sensory SCs express *atoh1a* to become HC precursors. Based on *atoh1a* in situ results (Figure 2C) 6.42%±0.68% SCs express *atoh1a* in 5dpf larvae. Based on our new analysis with *atoh1a:dTomato* reporter in Figure 5—figure supplement 1D, 2.44%±1.03% SCs have dTomato expression in normal condition. However, data in Figure 5—figure supplement 1C-D showed that the ratio of dTomato^+^ SCs significantly increased to 41.78%± 8.47% in LY+neo. In addition, many MCs (49.05%±11.32%), which never express dTomato in 5dpf larvae and neo, were labeled by dTomato in LY+neo. These data suggest that *atoh1a*^+^ cells were converted into multipotential progenitors in severe injury.

However, the lineage tracing experiment using cre/loxP system is necessary to confirm whether *atoh1a*^+^ cells are dedifferentiated or not. Since we were not able to do this experiment because of technology reason, we have already toned down our conclusions about dedifferentiation in our manuscript.

Their sentence "powerful capacity to regenerate HCs sustains after multiple rounds of damage…" references Cruz et al., 2015. This in one of two concurrent papers showing regeneration after multiple rounds of hair-cell ablation. The second is Pinto-Teixeira et al., 2015, which should be cited as well.

Thank the reviewer for the advice. We have added the other reference in the manuscript.

They state that the above papers "indicates that a population of stem/progenitor cell pool exists to self-renew and regenerate." However, there is ample evidence that the entire supporting cell population is able to convert to a stem cell-like is necessary. In other words, a stem cell population as a persistent cellular state has not been demonstrated.

The reviewer is correct that SCs are able to convert to a stem cell-like population in regeneration. Our results also indicate that *atoh1a*^+^ HC precursors gained sox2 expression to promote regeneration post severe injury. We have deleted the sentence in the text to avoid the confusion.

"Sox2 marks the proliferative progenitors that produce both SCs and HCs in homeostatic and regenerative NMs (21)."This is true, but only because sox2 is expressed by nearly every non-sensory cell. Because sensory cells are postmitotic, their conclusion is obvious.

The reviewer is correct that most SCs have sox2 expression in normal injury. Counting results (Author response image 1) show that about 60% of SCs or MCs are sox2^+^ in 5dpf normal and neo-treated larvae.

**Author response image 1. sa2fig1:** Some SCs and MCs in neuromast are negative with sox2 expression. (**A**)*ET4;ET20* Larvae were treated with neo and immediately collected for sox2 immunostaining. The white arrowheads pointed out SCs and MCs that were sox2 negative. (**B**) Total number of SCs or MCs and sox2^+^ ones were counted, and sox2^+^ ratios were calculated. About 60% of SCs or MCs express sox2.

"These results indicate that sox2^+^ progenitors in neuromast have high potential to restore themselves when exhausted by severe injury."Again, true and obvious because sox2 is expressed by nearly every non-sensory cell.

I agree with the reviewer that most SCs have sox2 expression in normal injury. But the sox2 expression pattern has changed in our new severe-injury model. We found that very few sox2^+^ cells (2.83±0.61, LY+neo group, Author response image 2, also in original Figure 1B) survived. But many *atoh1a*^+^ cells (11.8±0.78, LY+neo group in Author response image 2, related with Figure 2C) were still alive and became the main population post severe injury (Figure 2C).

**Author response image 2. sa2fig2:** Sox2^+^ progenitors were eliminated and *atoh1a^+^* HC precursors dominated post severe injury. (A) Larvae were treated with LY, neo or LY+neo and collected immediately after neo treatment for sox2 immunostaining. Sox2^+^ cell numbers were counted, and the results showed that very few sox2^+^ cells survived post LY+neo. (B) Larvae were treated with LY, neo or LY+neo and collected immediately after neo treatment for *atoh1a* RNA probe hybridization. Results showed that significantly higher number of *atoh1a*^+^ cells survived post LY+neo.

"These results indicate that intensive proliferation is necessary to facilitate regeneration post severe injury."This has been shown by no less than 10 previous papers already.

All the published papers used neo-treated model to analyze proliferation post normal injury. We compared LY+neo versus neo for EdU incorporation, and found that proliferation in SCs or MCs is highly increased (Figure 1—figure supplement 2). Therefore, we draw the conclusion that intensive proliferation facilitates regeneration post severe injury. Our finding is novel since this is the first study testing proliferation post severe injury.

Nuclear localized yap was inhibited when atoh1a is inhibited with hs:notch1a, which suggests that yap is activated in atoh1a^+^ committed HCs.This is puzzling because hair cells are postmitotic. In addition, is nuclear localization of yap takes place in atoh1a^+^ cells and activated Notch blocks the production of these cells, it is only obvious thet the authors will see no nuclear localized yap.

We have collected LY+neo-treated *hs:notch1a* larvae to check yap immunostaining. Our data (Figure 2F) showed that *atoh1a*^+^ cells were eliminated in *hs:notch1a*. And data in updated Figure 2—figure supplement 2E showed that nuclear-localized yap was inhibited in *hs:notch1a*. Therefore, we concluded that yap is activated in *atoh1a*^+^ cells post LY+neo.

"Taken together, we found that yap is highly activated by severe injury and binds directly to lin28a promoter to initiate its transcription in atoh1a^+^ committed HCs."I could not find direct evidence that yap binds lin28a promoter. It would help me if the authors pointing the data supporting the claim.

In Figure 2H, we used yap-immunoprecipitated DNA extracts for PCR (CHIP-PCR) using primers against *lin28a* promoter. We added the location of PCR primers in the updated figure. Results showed that PCR product against *lin28a* promoter was detected post LY+neo but not in LY. This piece of data indicates that yap binds lin28a promoter post severe injury.

"We found that the regenerated HC number was decreased by lin28a deficiency (Figure 3A)."What I see is that the entire neuromast was nearly half the size. Reduced hair-cell numbers may be an indirect consequence of lower proliferation, cellular extrusion or increased cell death.

Data in Figure 3A showed that HC number was decreased in *lin28a* mutant. The reviewer is correct that the HC decrease may be caused by loss of SC proliferation or other possible reasons. We didn’t do further analysis since we mainly focus on progenitor proliferation rather than HC production in this manuscript. But thank the reviewer for pointing out other possibilities which we will further analyze in the future.

"The proliferative SCs are localized in dorsal and ventral poles of neuromast in homeostasis and neo-induced regeneration"They reference Romero-Carvajal et al., 2015. Yet, earlier reports of the patterns of supporting-cell proliferation are: Ma et al., 2008, and Wibowo et al., 2011.

Thank the reviewer for the suggestion. We have added the other two references in the manuscript.

"The atoh1a:dTomato reporter labeled a part of but not all ET4 positive mature HCs in normal condition, indicating that the reporter is very mosaic (Figure 5C, ctrl)."An alternative interpretation, supported by findings of Wibowo et al., 2011, is that atoh1a:dTomato is expressed by supporting cells that may not advance to the hair-cell progenitor or hair-cell fates.

Thank you to the reviewer for this comment. I learned from the published paper by Wibowo et al., 2011 that young HCs are dTomato^+^ cells and mature HCs lose dTomato expression. Therefore, the mosaic expression of dTomato in HCs in Figure 5C may be caused by different states of HC maturity. We have changed the description accordingly in the manuscript.

We found a few dTomato^+^ SCs in normal and neo conditions (Figure 5—figure supplement 1) and the number is highly increased post LY+neo.

"However, more than half of the dTomato^+^ cells turned on sox2 expression at 12h and 48h post LY+neo and become dTomato^+^sox2^+^ cells (Figure 5D), suggesting that atoh1a^+^ cells were converted to sox2^+^ progenitors upon severe injury."Again, because sox2 is expressed by nearly every non-sensory cell, it is not surprising that some cells will co-express atoh1a and sox2.

I agree with the reviewer that most SCs have sox2 expression in normal larvae. Atoh1 is expressed in differentiating HC precursors and our data showed that very few dTomato^+^sox2^+^ cells existed in normal neuromast (updated Figure 5A-D). In accordance, several publications showed that atoh1 and sox2 is antagonistic in development (Dabdoub et al., 2008, Zhang et al., 2017). But the number of dTomato^+^sox2^+^ cells was highly increased post LY+neo, indicating that *atoh1a*^+^ cells gained sox2 expression in severe injury.

"However, it's only in LY+neo-induced severe injury that dTomato^+^ cells converted to sox2^+^ cells labeled by sox2:GFP (white dots in Figure 5F and Video 3), which confirms the dedifferentiation of atoh1a^+^ cells into sox2^+^ progenitors. „This statement is not supported by evidence presented in the manuscript.

Our data in updated Figure 5—figure supplement 1B and Video 3 showed that the two blue-dotted dTomato^+^ cells were GFP negative in the beginning, and divided into four daughter cells which gained GFP expression. We have changed the conclusion about dedifferentiation to that *atoh1a*^+^ cells gained sox2 expression in severe injury. Thank the reviewer for pointing out this concern.

Reviewer #3:[…]1) The hypothesis and the evidence point to lin28a triggering down-regulation of atoh1a and a return to a support cell fate. It should therefore be true that in the earlier developing larvae, if you induced lin28a expression ectopically, you should be able to inhibit natural differentiation of hair cells.

Thank you to the reviewer for the suggestion. To address the question whether *lin28a* affect early differentiation, we have analyzed 3dpf and 5dpf *hs:lin28a* larvae for HC counting. The results (updated Figure 4—figure supplement 1) showed that *lin28a* has no effect on HC differentiation during development. Our data in Figure 3 and Figure 4 also indicate that *lin28a* has no effect on HC differentiation (HC+, EdU incorporated HCs). And data in updated Figure 6—figure supplement 1 showed that differentiation related gene *her4* was not changed in *hs:lin28a*. All these data indicate that lin28a has no effect on differentiation in either development or regeneration. It could be possible that new *atoh1a*^+^ HC precursors are produced when *lin28a* inhibits the differentiation of old *atoh1a*^+^ cells.

Part of the model presented is that lin28a dedifferentiated cells contribute to new supporting cell populations too. Is it possible that the lin28a cells can only go "backwards" far enough to divide but not far enough to become support cells again? Under those circumstances the new support cells could come from the mantle cells which have been shown to be capable of fully regenerating all cell types in the neuromast. If it is really complete dedifferentiation, there should be more supporting cells in these larvae and not cells that are neither HC nor SC. It would be interesting if there could be lineage tracing of "dedifferentiated" atoh1^+^ cells. Can they actually become supporting cells?

Our new data in updated Figure 5—figure supplement 1C-D showed that *atoh1a* label mostly HCs and a small number of SCs in normal and neo-treated larvae. Higher number of *atoh1a*^+^ SCs was detected in LY+neo compared with neo. More importantly, many MCs were also labeled with *atoh1a* post LY+neo, which was not detected in normal or neo-treated conditions. This piece of data suggests that *atoh1a*^+^ cells behaved like progenitor/stem cells that give rise to many support cell types post severe injury. However, the lineage tracing experiment using cre/loxP system is necessary to confirm whether *atoh1a*^+^ cells are dedifferentiated or not. Since we were not able to do this experiment because of technology reason, we have already toned down the conclusions in our manuscript. Now the dedifferentiation conclusion was changed to that *atoh1a*^+^ HC precursors gained sox2 expression post severe injury.

[Editors’ note: what follows is the authors’ response to the second round of review.]

Essential revisions:1) Improve the Western blots to strengthen the data here. At the moment, the immunostainings are not clear, and the increase in yap1 is not clear on the Western blot. Indicate Molecular Weight on the blot.2) Confirm the specificity of your antibodies by checking on yap1 mutants.3) Confirm the specificity of the Verteporfin inhibitor by comparing to the phenotype of yap1 mutants.4) Please attend to all the listed minor revisions and, where possible, to any other major revisions, such as citation of additional studies in the Introduction (or provide a rebuttal of these points).

Thank you for providing us the opportunity for revision. We also appreciate the time and effort of the two reviewers. Based on the reviewers’ valuable comments we have performed a series of new experiments that were briefly summarized below.

1) We have optimized the western-blot conditions to detect yap expression post severe injury. The new results clearly showed that yap is upregulated at 1h and 5h post LY+neo.

2) We have used yap mutants to check the specificity of yap antibody by western-blot and immunostaining. The results showed that the yap antibody is very specific for detecting yap protein in siblings but not yap mutants.

3) We have used yap mutants to confirm that yap is required for the severe-injury-induced regeneration.

4) We also provided evidence that the classic target genes of yap such as cyr61 and ctgfa are upregulated post severe injury, which is inhibited by verteporfin.

Reviewer #2:The study reveals an interesting, new mechanism driving neuromast regeneration upon severe injury. While I think the study is potentially interesting for eLife readership, I feel some parts are not as solid as others, in particular the part on the role of yap1. Please see my specific comments below.1) The Introduction is short and miss to refer to some important studies that have been published on similar questions including, for example, work from the Lopez-Shier lab (PintoTeixeira et al., 2015, ViaderLlargues et al., 2018, for example)

Thanks to the reviewer for the nice suggestion. We have previously included PintoTeixeira et al., 2015 in the Introduction. Now we added ViaderLlargues et al., 2018 in the Introduction as an important reference for the high potential of SCs to regenerate after severe damage.

2) Using hs:notch1a to block Atoh1a function is not ideal. The authors should use atoh1a mutants which are available or Morpholino which have been published.

In Figure 2F we found that LY+neo-induced *lin28a* is decreased in *hs:notch1a*. I agree with the reviewer that *atoh1a* mutants or *atoh1a* morpholino is more specific to block *atoh1a* function than *hs:notch1a*. However, HCs are absent in *atoh1a* mutants or morphants, which makes it impossible to cause severe damage by combined treatment of LY and neomycin. In order to induce severe damage, we have to use *hs:notch1a* to transiently block *atoh1a* expression without affecting hair cell development.

The aim of this experiment is to check whether *lin28a* is induced in *atoh1a* expressing cells. In addition to this piece of data, we observed that *lin28a* induction is increased in *hs:atoh1a* and *lin28a* is co-labeled with *atoh1a* post severe injury. All these evidences clearly indicate that *lin28a* is induced in *atoh1a*^+^ cells post severe injury.

3) The authors should demonstrate the specificity of the yap1 and taz antibodies they are using by showing the absence of signal in respective mutants (available). These controls should be shown both for western-blot and immunostaining.

Our new results in updated Figure 2—figure supplement 2F showed that yap antibody had good immunostaining signal in LY+neo-treated neuromast, which was lack in yap mutants. Western-blot results in Figure 2—figure supplement 2C have detected the yap band in siblings but not in yap mutants. These results indicate that yap antibody is specific in recognizing yap protein.

4) On the western-blot in Figure 2—figure supplement 1B, taz seems to be much more upregulated than yap1. Why do the authors then focus exclusively on yap1?

We have optimized the western-blot conditions with yap antibody and updated the Figure 2—figure supplement 2B results. We found that yap protein level was upregulated at 1h and 5h post LY+neo while taz was upregulated until 5h. In addition, yap but not taz was detected in neuromast by immunostaining at 1h post LY+neo (updated Figure 2—figure supplement 2D-G and data not shown). Based on these results, yap is activated earlier than taz post severe injury and might be responsible for regeneration. Indeed, we found that the regenerative proliferation and *lin28a* induction were blocked in yap mutants (Figure 3F and 2J). All these indicate that yap activation is required for severe-injury-induced regeneration.

5) The yap signal and the difference between control and treated in Figure 4C and 4E are not convincing.

I assume the reviewer referred to Figure 2—figure supplement 2C and E in the original figures which were now updated as Figure 2—figure supplement 2D and Figure 2—figure supplement 2G. We have proved the specificity of yap antibody by using yap mutants in the updated Figures (details in major point 3). The signal of yap immunostaining seemed localized in both nucleus and cytoplasm of neuromast cells, possibly because yap shuttles between nuclei and cytoplasm. We also noticed that the injury induced yap expression not only in neuromast but also in surrounding skin and muscle. Only yap expressing cells within the neuromasts were chosen and divided by total cell number to calculate the ratio. The ratio of yap expression was significantly increased post LY+neo, which was inhibited in *hs:notch1a*.

6) "The induction of yap and mst2 was blocked by verteporfin, an inhibitor of yap-mediated transcription by blocking yap-tead1 interaction": why are the expression of yap and mst2 blocked upon inhibition of yap-mediated transcription? Has it previously been shown that yap and mst2 are transcriptional targets of yap? The authors should look at classical yap/taz transcriptional targets.

Thanks to the reviewer for the suggestion. We have tested the classic yap transcriptional target genes cyr61 and ctgfa post severe injury by real time PCR. Results in updated Figure 2—figure supplement 2H indicated that expression levels of cyp61 and ctgfa were increased post severe injury, which was inhibited by verteporfin. This indicates that yap activity is activated post severe injury. Our results in updated Figure 2—figure supplement 2I suggested that yap probably upregulated expressions of itself and mst2 to create a positive feedback loop.

7) Several yap and taz zebrafish mutants have been published. The authors should perform experiments in these mutants instead of – or in addition to – Verteporfin experiments.

Thanks to the reviewer for the suggestion. We have tested regenerative proliferation in yap mutants. The proliferation was significantly decreased in yap mutants (updated Figure 3F), which resembled the phenotype of verteporfin treatment (updated Figure 3G).

8) "We used CHIP-PCR to verify that yap binds to a region 100bp downstream of lin28a transcription start site". How did the author choose this region?

We used the promoter region of *lin28a* to search for the conserved yap binding motif by sequence similarity. There is a yap binding motif located at 100bp downstream of *lin28a* transcription start site, where a tead1 binding motif is located nearby. This suggested that yap might cooperate with tead1 to activate *lin28a* transcription. We verified the binding of yap to this region by CHIP-PCR (Figure 2H).

9) Figure 3A: does the Y-axis indicate EdU cell number (as indicated on the figure) or absolute numbers of HS, SC and MC (as the text suggests)?

Thanks to the reviewer for pointing out this typo. It should be the absolute number of HC, SC and MC, and we changed the Y-axis label accordingly.

10) "The EdU positive SC cells that were located in each quadrant with no polarization in sibling post LY+neo were almost cleared in lin28a mutant (Figure 3C-D)": The link to polarization all the sudden here is not clear.

Results in Figure 3D showed that the EdU+ proliferative cells post LY+neo is not polarized since the statistical analysis is not significant (N.S. at the bottom). Therefore the severe-injury-induced proliferation is not polarized, which is unlike the dorsal-ventral localization of EdU+ cells post neo (Figure 4C and 4D).

11) Figure 3F: the authors should use yap mutants in addition to Verteporfin treatment.

We have analyzed yap mutants for LY+neo-induced regeneration and added to updated Figure 3F. Please refer to point 7 for details.